# Learning to Reject with a Fixed Predictor: Application to Decontextualization

**Christopher Mohri[1], Daniel Andor[2], Eunsol Choi[3], Michael Collins[2], Anqi Mao[4], Yutao Zhong[4]**
[1]Stanford University, [2]Google, [3]The University of Texas at Austin, [4]Courant Institute
`xmohri@stanford.edu`, {`danielandor,mjcollins`}`@google.com`
`eunsol@@utexas.edu`, {`aqmao,yutao`}`@cims.nyu.edu`

## Abstract

We study the problem of classification with a reject option for a fixed predictor, crucial to natural language processing. We introduce a new problem formulation for this scenario, and an algorithm minimizing a new surrogate loss function. We provide a complete theoretical analysis of the surrogate loss function with a strong $\mathcal{H}$-consistency guarantee. For evaluation, we choose the *decontextualization* task, and provide a manually-labelled dataset of $2,000$ examples. Our algorithm significantly outperforms the baselines considered, with a ~$25\%$ improvement in coverage when halving the error rate, which is only ~$3\%$ away from the theoretical limit.

## 1 Introduction

Large language models, often trained with billions of parameters, have achieved impressive performance in recent years (Raffel et al., 2020) and are used in a wide variety of natural language generation tasks. However, their output is sometimes undesirable, with *hallucinated content* (Maynez et al., 2020; Filippova, 2020), and much work remains to fully understand their properties. In many applications, such as healthcare, question-answering systems, or customer service, incorrect predictions are particularly costly and must be avoided. This motivates the design of algorithms for large language models and other natural language processing (NLP) tasks that achieve high precision on a large fraction of the input set, while abstaining on the rest. How can we devise such accurate models that allow a reject option?

A common technique adopted in the past is that of *confidence-based models*, where rejection is defined by some threshold on the predictor's scores and admits a fixed cost (Hendrickx et al., 2021). Chow (1957; 1970) was the first to provide an analysis of the trade-off between error and rejection rate, as well as the associated Bayes optimal solution. The rejection rule was later studied based on the receiver operating characteristic (ROC) curve (Tortorella, 2000; Santos-Pereira and Pires, 2005; Pietraszek, 2007; Landgrebe et al., 2006). Some subsequent work has focused on minimizing surrogate loss functions for the cost-based objective, with various theoretical guarantees (Bartlett and Wegkamp, 2008; Grandvalet et al., 2008; Yuan and Wegkamp, 2010). In NLP, it has been reported that scores from popular large language models such as T5, BART, and GPT-2 are poorly calibrated (Jiang et al., 2020; Kumar and Sarawagi, 2019; Lewis et al., 2019; Raffel et al., 2020), similar to modern neural networks (Guo et al., 2017), but Xin et al. (2021) proposed a simple regularization trick during training to improve these scores. The method used by several other authors can also be viewed as an instance of confidence-based models (Kamath et al., 2020; Zhang et al., 2021; Garg and Moschitti, 2021; Dong et al., 2018; Varshney et al., 2022; Li et al., 2024; Chen et al., 2024). Here, the idea consists of first learning a scoring function defined over pairs $(x, y)$, and next applying the confidence-based technique to the scoring function.

However, as shown by Cortes et al. (2016)[see Figure 2], straightforward confidence-based methods are in general suboptimal. When the predictor learned is not the Bayes optimal solution, in general, a more complex rejection rule is needed to achieve better performance. The authors suggested instead seeking a suitable *rejector* out of a family of rejection functions that may be richer than that of confidence-based threshold ones. They gave theory and algorithms for learning the predictor and rejector simultaneously by minimizing a *rejection loss*, whose definition takes into account the cost

of rejection, $c$. As shown by Ni et al. (2019), extending this predictor-rejector framework to the multi-class setting is challenging and remains an open problem, which has been solved positively by Mao et al. (2024a;c) and extended to multi-expert deferral (Mao et al., 2023a; 2024b). Instead, Charoenphakdee et al. (2021) decomposed multi-class abstention into multiple one-versus-all binary classification problems from a cost-sensitive point of view. Recently, such cost-sensitive approach has been shown empirically to be inferior to the latest approach for abstention (Cao et al., 2022).

We aim to design accurate models with a rejection option for NLP generation tasks such as *decontextualization* (Choi et al., 2021). Decontextualization involves editing a sentence within a passage such that it can stand alone. It is critical in this task to make confident predictions, or abstain; if we are to edit an author's words, we must be sure to do so correctly. One solution would consist of adopting the rejection loss function of Cortes et al. (2016; 2023) and of learning, simultaneously, a predictor $f$ and a rejector $r$ that minimize the rejection loss. However, for NLP tasks such as decontextualization, this faces a key obstacle: the full set of *accurate outputs* for a given input sentence may be large and is typically not at the learner's disposal. For example, in decontextualization, some standard transformations such as passivation applied to an accurate output can immediately lead to a large number of other accurate outputs. To minimize the rejection loss, however, the learner must be able to evaluate the correctness of any potential predictor on any example in the training sample. But, apart from the single label in the training sample, other potential accurate labels are not provided and thus the correctness of a potential predictor cannot be checked. How can we then learn an accurate rejection-based model for such settings?

One way to proceed is instead to first learn a predictor $f$, using the standard techniques adopted for large language models, for example by minimizing the cross-entropy loss in next-token prediction. Next, given that predictor, it is not hard to manually assign a binary label to the output of $f$ on some relatively small set of held-out examples indicating their correctness. The problem of interest then consists of finding the best rejector function $r$ minimizing the rejection loss with $f$ fixed. Here, the rejector function $r$ takes the input $x$ together with the output $f(x)$ as its own input. We refer to this procedure as *learning to reject with a fixed predictor $f$*.

The resulting binary rejection loss function for $r$ is hard to optimize directly and, instead, we need to resort to a surrogate loss. Cortes et al. (2016) gave a consistent surrogate loss for their full rejection loss, which involved learning both $f$ and $r$. However, since $f$ is fixed in our context, we cannot benefit from the corresponding consistency guarantees. Instead, we first propose a parametric family of surrogate losses inspired by their work for learning $r$ alone. Next, we prove that these surrogate losses benefit from strong consistency results, when a suitable constraint holds for the parameters.

Our results make use of the recent work of Awasthi et al. (2022a), which gave general tools for deriving $\mathcal{H}$-*consistency bounds*. These are bounds relating directly the excess error or estimation error of the surrogate loss to those of the original loss (here the rejection loss). Thus, they are stronger guarantees than asymptotic Bayes-consistency results. Furthermore, they can be extended to other hypothesis sets $\mathcal{H}$ than that of the family of all measurable functions. We use those tools to prove the first $\mathcal{H}$-consistency bound for our surrogate rejection loss, which provides a strong justification for its adoption for tackling the problem of interest.

The rest of this paper is organized as follows. In Section 2, we formulate our learning problem, which consists of learning a rejector given a fixed predictor. In Section 3, we introduce confidence-based models and contrast them with our two-step model. In Section 4, we derive our surrogate rejection loss. In Section 5, we prove a strong $\mathcal{H}$-consistency bound for the proposed surrogate rejection loss. In Section 6, we give a description of the decontextualization task and the annotation procedure. Section 7 reports the experimental results of our surrogate rejection loss, and shows that it compares favorably with several baselines in a decontextualization task.

## 2 LEARNING PROBLEM

We consider the problem of *sequence-to-sequence modeling with high confidence* in natural language processing. The general objective is to design an algorithm that only returns an output when it is highly likely to be correct, while still guaranteeing a high coverage.

Let $\mathcal{X}$ denote the input and $\mathcal{Y}$ the output set of sequences and let $\mathcal{D}$ be a distribution over $\mathcal{X} \times \mathcal{Y}$. The problem can be formalized in terms of a sequence-to-sequence *predictor* $f : \mathcal{X} \to \mathcal{Y}$ and a *rejector*

$r: \mathcal{X} \times \mathcal{Y} \to \mathbb{R}$. A non-positive sign for the output of the rejector is interpreted as rejection, and a positive one as acceptance. Formally, on an input $x \in \mathcal{X}$, we have

$$(f, r)(x) = \begin{cases} f(x), & \text{if } r(x, f(x)) > 0 \\ \text{reject}, & \text{if } r(x, f(x)) \leq 0. \end{cases}$$

Given a hypothesis set $\mathcal{F}$ of sequence-to-sequence predictors and a family of rejectors $\mathcal{R}$, for a cost $c \in [0, 1]$, the natural target loss for the abstention problem can be formulated as follows:

$$L(f, r, x, y) = \mathbb{I}_{f(x) \neq y} \, \mathbb{I}_{r(x, f(x)) > 0} + c \, \mathbb{I}_{r(x, f(x)) \leq 0} \,. \tag{1}$$

We wish to minimize the error $\left(\mathbb{I}_{f(x) \neq y}\right)$ on accepted examples $\left(\mathbb{I}_{r(x, f(x)) > 0}\right)$, while imposing a fixed cost $c$ on rejected examples. As the cost of rejection increases, one can expect a higher coverage, but lower precision. One difference with respect to the framework of Cortes et al. (2016; 2023) is that, here, the rejector takes as argument the prediction as well. More generally, as already pointed out in that previous work, the rejection cost $c$ can be a function of $x$ and $f(x)$, not just a constant. The problem of selective classification (Gelbhart and El-Yaniv, 2019), which is based on the choice of a threshold function, can be viewed as a special case of this framework.

**Distribution model**. Before describing any method for tackling this problem, we wish to discuss the distributional model adopted in certain NLP tasks such as decontextualization. In standard learning tasks, there may be multiple correct labels $y$ for the same input $x$ and with an i.i.d. training sample, we expect to come across all of these labels with their conditional probabilities given $x$.

In complex NLP tasks such as decontextualization, however, there may be a relatively large number of correct $y$s for a given $x$: the task is highly non-deterministic. As discussed, some standard transformations such as passivation applied to one $y$ can immediately lead to a much larger number of correct output sentences. Thus, it is not realistic to demand from labelers to supply all possible correct sentences $y$ for a given input $x$. On the other hand, we do not wish to consider it to be an error if a model returns a desirable sentence that does not exactly match any of the labels provided by the labelers.

What should be the correct distributional model to adopt for such NLP tasks? We can consider two models: a *deterministic model* where only a single label is accepted as correct for any input sentence; or a more general and more useful *stochastic* or *non-deterministic model* where, in addition to the single label $y$ (or few labels) provided by labelers, any other label $y'$ returned by the model is viewed as correct provided that $y'$ is sufficiently similar to $y$, based on some pre-defined similarity measure. It is important to note that this pre-defined similarity measure may be difficult to specify, and may therefore require expert human annotation.

Adopting the non-deterministic model, for $(x, y) \sim \mathcal{D}$ and a given $f \in \mathcal{F}$, we amend our indicator function $\mathbb{I}_{f(x) \neq y}$ measuring incorrectness in (1). Instead, we measure $\mathbb{I}_{f(x) \notin A_{x,y}}$, where $A_{x,y} \subseteq \mathcal{Y}$ implements some similarity measure and describes a set of acceptable outputs $y$. Provided that we have a boolean random variable $a \in \{-1, +1\}$ derived from $(x, y)$ and $f(x)$ indicating membership to $A_{x,y}$, the event where $f(x)$ is an acceptable output, we can simplify this indicator function to $\mathbb{I}_{a=-1}$. Since $f$ is fixed, we remove it as an argument to $r$, and the distribution over $(x, y) \sim \mathcal{D}$ leads to a distribution $(x, a) \sim \overline{\mathcal{D}}$ induced by $f$. We will refer to the following as the *induced rejection loss* defined for any rejector $r$ and pair $(x, a)$:

$$\ell(r, x, a) = \mathbb{I}_{a=-1} \, \mathbb{I}_{r(x) > 0} + c \, \mathbb{I}_{r(x) \leq 0} \,. \tag{2}$$

In the following, we will distinguish two methods for learning the rejection function $r$: the so-called *confidence-based method* where $r$ is simply defined as a threshold based on the predictor $f$, and a *two-step learning method* where first $f$ is learned and then $r$.

## 3 LEARNING METHODS

In this section, we describe in more detail the two learning methods previously mentioned.

### 3.1 CONFIDENCE-BASED METHOD

The confidence-based method is perhaps the most commonly used one to define a rejection function. Let $f(x) = \operatorname{argmax}_y s(x, y)$ for some scoring function $s : \mathcal{X} \times \mathcal{Y} \mapsto \mathcal{R}$, which could be $p(y|x; \omega)$,

where $\omega$ represents model parameters. Then, a threshold value $\theta$ is set based on some function of the scores $s(x, y)$ assigned to each $y \in \mathcal{Y}$ for a given $x \in \mathcal{X}$. If $s_i$ and $s_j$ are the final scores assigned to $(x_i, y_i) \sim \mathcal{D}$ and $(x_j, y_j) \sim \mathcal{D}$ respectively, we wish to have *monotonicity*: $s_i \geq s_j \Leftrightarrow \mathbb{I}_{f(x_i) \neq y_i} \leq \mathbb{I}_{f(x_j) \neq y_j}$ (Geifman and El-Yaniv, 2017).

The *MaxProb* method is simply defined in terms of the highest score (Hendrycks and Gimpel, 2016). In that case, the rejection function is a function mapping from $\mathcal{X}$ to $\mathbb{R}$ defined by $r(x) = s(x, f(x)) - \theta$, for some choice of the threshold $\theta \in \mathbb{R}$. Another popular scoring function is based on Monte-Carlo dropout (Smith and Gal, 2018; Gal and Ghahramani, 2016), measuring statistics such as the mean or negative variance of the scores $s(x, y)$. Fitting the threshold to guarantee that it matches a target precision has been studied in (Geifman and El-Yaniv, 2017). In our experiments, we follow a simpler procedure, as this is not our focus; we are interested in the quality of the underlying scores.

## 3.2 Two-step method

In the two-step method, a predictor $f$ is first learned by minimizing a surrogate loss for $\mathbb{I}_{f(x) \neq y}$, such as the cross-entropy loss. Next, the rejection function $r$ is learned as a binary classifier.

Note that, to learn $r$ in the non-deterministic distributional model requires binary labels for pairs $(x, f(x))$ indicating if the output sequence $f(x)$ is indeed a good label for $x$, or formally, if $f(x) \in A_{x,y}$. As already discussed, that information cannot be directly derived from the label $y$ of $x$ in the training sample since the correct labels are typically not unique in NLP tasks. One way to derive that information is to manually label such pairs. This admits two disadvantages: the manually assigned labels are specific to the predictor $f$ previously learned and thus cannot be reused for different predictors; and of course this requires manual labeling which is typically costly. However, if the cost is not too significant, one can label a moderately-sized dataset and then train a classifier on it.

One approach for training such a classifier is to use the standard cross-entropy loss function. More specifically, for pairs $((x, f(x)), a)$, where $a$ represents the label or annotation, one can train with direct supervision using the binary cross-entropy loss. However, similar to the *MaxProb* method, this also requires setting a threshold $\theta$ for acceptance based on model scores. One can only hope that the classifier produces higher-quality scores, where quality is associated with monotonicity. Thus, both methods are based on straightforward threshold rejection. Additionally, to the best of our knowledge, minimizing the cross-entropy loss does not have any proven guarantee with respect to our main objective: minimizing the induced rejection loss. In the next section, we tackle both of these problems: we introduce a new loss function with a built-in threshold that *directly minimizes* the induced rejection loss.

In Appendix C, we briefly discuss a comparison to a cost-sensitive classification formulation. This solution is reported to be inferior to the state-of-the-art empirically (Cao et al., 2022) and is not proven to benefit from an $\mathcal{H}$-consistency bound, unlike the solution we present in the next sections.

## 4 Surrogate rejection loss

Cortes et al. (2016; 2023) study the joint optimization problem in (1) where the predictor $f$ is a binary classifier, and define a convex surrogate loss upper-bounding their rejection loss $L$. We use the same technique to upper-bound the induced rejection loss. Specifically, the following inequality holds, where $\alpha$ and $\beta$ are positive parameters and $x \mapsto \phi(-x)$ and $x \mapsto \psi(-x)$ are convex functions upper-bounding $\mathbb{I}_{x \leq 0}$:

$$\ell(r, x, a) = \mathbb{I}_{a=-1} \mathbb{I}_{r(x)>0} + c \mathbb{I}_{r(x) \leq 0} = \mathbb{I}_{a \leq 0} \mathbb{I}_{r(x)>0} + c \mathbb{I}_{r(x) \leq 0}$$

$$\leq \max\{\mathbb{I}_{a \leq 0} \mathbb{I}_{-r(x)<0}, c \mathbb{I}_{r(x) \leq 0}\} \leq \max\{\mathbb{I}_{\max(a, -r(x)) \leq 0}, c \mathbb{I}_{r(x) \leq 0}\}.$$

Next, since the maximum is lower-bounded by the average, we can write:

$$\ell(r, x, a) \leq \max\{\mathbb{I}_{\frac{a-r(x)}{2} \leq 0}, c \mathbb{I}_{r(x) \leq 0}\} = \max\{\mathbb{I}_{\alpha \frac{a-r(x)}{2} \leq 0}, c \mathbb{I}_{\beta r(x) \leq 0}\}$$

$$\leq \max\{\phi\left(\tfrac{\alpha}{2}(r(x) - a)\right), c \psi(-\beta r(x))\} \leq \phi\left(\tfrac{\alpha}{2}(r(x) - a)\right) + c \psi(-\beta r(x)).$$

We use the exponential function for $\phi$ and $\psi$, giving our surrogate rejection loss function:

$$\ell(r, x, a) \leq e^{\frac{\alpha}{2}[r(x)-a]} + ce^{-\beta r(x)}.$$

While the induced rejection loss $\ell(r, x, a)$ is provably NP-hard to optimize, our surrogate loss is convex and differentiable. A key insight is that models optimized with our loss function have a built-in threshold of 0; they are directly optimized for a *specific* precision. Thus, there is no need to further specify some score threshold as in all methods previously described. In those methods, one can still target certain precision levels through the choice of threshold, but the *underlying scores* are not necessarily favorable for that precision level.

While Cortes et al. (2016; 2023) proved theoretical guarantees for a joint minimization of their loss function in their binary setting, these do not naturally apply to our problem (the predictor usually does not have zero error and is not the Bayes predictor). In the next section, we prove strong theoretical guarantees for minimizing our surrogate loss function.

## 5 $\mathcal{H}$-CONSISTENCY BOUND

In this section we prove $\mathcal{H}$-consistency bounds, a concept introduced by Awasthi et al. (2022a;b), of our surrogate rejection loss function with respect to the induced rejection loss. To the best of our knowledge, these non-asymptotic bounds are the strongest guarantees known regarding the minimization of surrogate loss functions (Awasthi et al., 2021a;b; 2023; 2024; Mao et al., 2023b;d;e;c;f). We first introduce some basic concepts and adopt the notation of Awasthi et al. (2022a).

### 5.1 PRELIMINARIES

Let $\mathcal{X}$ denote the input space, $\mathcal{Y} = \{-1, +1\}$ the binary label space, and $\mathcal{D}$ a distribution over $\mathcal{X} \times \mathcal{Y}$. Let $\mathcal{R}$ denote a family of rejection functions mapping from $\mathcal{X}$ to $\mathbb{R}$. Then, the *generalization error* $R_\ell(r)$ and *minimal generalization error* $R^*_{\ell,\mathcal{R}}$ for a loss function $\ell(r, x, y)$ are defined by

$$R_\ell(r) = \mathop{\mathbb{E}}_{(x,y)\sim\mathcal{D}}[\ell(r, x, y)] \text{ and } R^*_{\ell,\mathcal{R}} = \inf_{r\in\mathcal{R}} R_\ell(r).$$

We will adopt the standard notation for the conditional distribution of $Y = 1$ given $X = x$: $\eta(x) = \mathcal{D}(Y = 1 \mid X = x)$. The generalization error can be expressed as $R_\ell(r) = \mathbb{E}_X[\mathcal{C}_\ell(r, x)]$, where $\mathcal{C}_\ell(r, x)$ is the *conditional $\ell$-risk* defined by $\mathcal{C}_\ell(r, x) = \eta(x)\ell(r, x, +1) + (1 - \eta(x))\ell(r, x, -1)$. The *minimal conditional $\ell$-risk* is denoted by $\mathcal{C}^*_{\ell,\mathcal{R}}(x) = \inf_{r\in\mathcal{R}} \mathcal{C}_\ell(r, x)$. We also use the following shorthand for the gap $\Delta\mathcal{C}_\ell(r, x) = \mathcal{C}_\ell(r, x) - \mathcal{C}^*_{\ell,\mathcal{R}}(x)$ and refer to it as *calibration gap*.

A key quantity that appears in their bounds is the $(\ell, \mathcal{R})$-*minimizability gap* $\mathcal{M}_{\ell,\mathcal{R}}$, which is the difference of the best-in class error and the expectation of the minimal conditional $\ell$-risk:

$$\mathcal{M}_{\ell,\mathcal{R}} = R^*_{\ell,\mathcal{R}} - \mathop{\mathbb{E}}_X[\mathcal{C}^*_{\ell,\mathcal{R}}(x)].$$

This is an inherent property of the hypothesis set $\mathcal{R}$ and distribution $\mathcal{D}$ that we cannot hope to estimate or minimize. As discussed later, the minimizability gap is zero when $\mathcal{R}$ is the family of all measurable functions $\mathcal{R}_{\text{all}}$. For simplicity, we omit dependency on $\mathcal{R}$ in all notation when $\mathcal{R} = \mathcal{R}_{\text{all}}$.

### 5.2 DEFINITION OF THE LOSSES AND THE DESIRED GUARANTEE

We consider the induced rejection loss function $\ell_2 = \ell$ defined for any rejection function or *rejector* $r: \mathcal{X} \to \mathbb{R}$ and $(x, a) \in \mathcal{X} \times \{-1, +1\}$ by $\ell_2(r, x, a) = \mathbb{I}_{a=-1}\mathbb{I}_{r(x)>0} + c\mathbb{I}_{r(x)\leq0}$. We will consider a surrogate loss function $\ell_1$ parameterized by $\alpha, \beta > 0$ and defined for any rejection function or *rejector* $r: \mathcal{X} \to \mathbb{R}$ and $(x, a) \in \mathcal{X} \times \{-1, +1\}$ by $\ell_1(r, x, a) = e^{\frac{\alpha}{2}[r(x)-a]} + ce^{-\beta r(x)}$. We will prove $\mathcal{H}$-consistency bounds for the surrogate loss $\ell_1$ with respect to the target loss $\ell_2$, when $r$ is in the family of all measurable functions $\mathcal{R}_{\text{all}}$. These are excess error bounds of the form $R_{\ell_2}(r) - R^*_{\ell_2} \leq f(R_{\ell_1}(r) - R^*_{\ell_1})$ valid for all $r$ for an increasing function $f$. To do so, we will use the following general theorem from (Awasthi et al., 2022a).

**Theorem 1.** *Assume that there exists a convex function $\Psi: \mathbb{R}_+ \to \mathbb{R}$ with $\Psi(0) = 0$ such that the following holds for all $r \in \mathcal{R}$ and $x \in \mathcal{X}$: $\Psi(\Delta\mathcal{C}_{\ell_2}(r, x)) \leq \Delta\mathcal{C}_{\ell_1}(r, x)$. Then, the following inequality holds for any $r \in \mathcal{R}$:*

$$\Psi(R_{\ell_2}(r) - R^*_{\ell_2,\mathcal{R}} + \mathcal{M}_{\ell_2,\mathcal{R}}) \leq R_{\ell_1}(r) - R^*_{\ell_1,\mathcal{R}} + \mathcal{M}_{\ell_1,\mathcal{R}}.$$

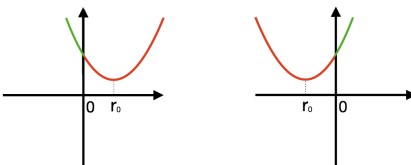

Figure 1: Lower bound for $\Delta\mathcal{C}_{\ell_1}$. The green denotes the values $r(x)$ can take. **Left:** $r(x) \leq 0$; **Right:** $r(x) \geq 0$. In both cases, the infimum is attained at $r(x) = 0$.

As shown by Steinwart (2007, Lemma 2.5), since $\ell_1(r, x, a)$ and $\ell_2(r, x, a)$ can be expressed in terms of $r(x)$ and $a$ alone, both minimizability gaps $\mathcal{M}_{\ell_1, \mathcal{R}_{\mathrm{all}}}$ and $\mathcal{M}_{\ell_2, \mathcal{R}_{\mathrm{all}}}$ vanish ($\mathcal{M}_{\ell_1, \mathcal{R}_{\mathrm{all}}} = \mathcal{M}_{\ell_2, \mathcal{R}_{\mathrm{all}}} = 0$). To make use of this theorem, in the next sections, we will derive the expression of the calibration gaps $\Delta\mathcal{C}_{\ell_2}$ and $\Delta\mathcal{C}_{\ell_1}$.

## 5.3 CALIBRATION GAPS

The following gives the expression of the calibration gaps. The proofs for Lemma 2 and Lemma 3 are deferred to Appendix A.1 and Appendix A.2 respectively.

**Lemma 2.** *The Bayes solution $r^*$ for the rejection loss can be expressed for all $x \in \mathcal{X}$ by $r^*(x) = \eta(x) - (1 - c)$. The calibration gap for the rejection loss is given for any $r \in \mathcal{R}_{\mathrm{all}}$ and $x \in \mathcal{X}$ by*

$$\Delta\mathcal{C}_{\ell_2}(r, x) = |\eta(x) - (1 - c)| \mathbb{I}_{r(x)r^*(x) \leq 0}.$$

**Lemma 3.** *Let $I_\eta(x)$ be defined by $I_\eta(x) = \eta(x)e^{-\frac{\alpha}{2}} + (1 - \eta(x))e^{\frac{\alpha}{2}}$ and define $\gamma$ by $\gamma = \frac{\alpha}{\alpha + 2\beta}$. Then, the calibration gap for the surrogate loss is given for any $r \in \mathcal{R}_{\mathrm{all}}$ and $x \in \mathcal{X}$ by*

$$\Delta\mathcal{C}_{\ell_1}(r, x) = e^{\frac{\alpha}{2}r(x)}I_\eta(x) + ce^{-\beta r(x)} - \frac{1}{1 - \gamma}\left(\frac{2\beta c}{\alpha}\right)^\gamma I_\eta(x)^{1-\gamma}.$$

## 5.4 MAIN RESULT

In this section, we present our main result. A key challenge in finding a function $\Psi$ relating the two calibration gaps is that $\Delta\mathcal{C}_{\ell_1}$ depends on the value $r(x) \in \mathbb{R}$, while $\Delta\mathcal{C}_{\ell_2}$ only depends on the sign of $r(x)$, via that of $r^*(x)$. The following provides a key solution to this problem.

**Proposition 4.** *Assume that there exists a convex function $\Psi: \mathbb{R}_+ \to \mathbb{R}$ with $\Psi(0) = 0$ such that the following holds for all $r \in \mathcal{R}_{\mathrm{all}}$ and $x \in \mathcal{X}$: $\Psi\big(|\eta(x) - (1 - c)| \mathbb{I}_{r(x)r^*(x) \leq 0}\big) \leq \Delta\mathcal{C}_{\ell_1}(0, x)$. Let $\bar{I}_c$ be defined by $\bar{I}_c = ce^{\frac{\alpha}{2}} + (1 - c)e^{-\frac{\alpha}{2}}$ and assume that $\frac{2\beta c}{\alpha} = \bar{I}_c$. Then, for any $r \in \mathcal{R}_{\mathrm{all}}$:*

$$\Psi\big(R_{\ell_2}(r) - R_{\ell_2}^*\big) \leq R_{\ell_1}(r) - R_{\ell_1}^*. \tag{3}$$

The proof is presented in Appendix A.3. The result shows that, instead, we only need to find a function $\Psi$ relating the gap $\Delta\mathcal{C}_{\ell_2}$ to $\Delta\mathcal{C}_{\ell_1}(0, x)$, which is no longer a quantity depending on $r(x)$. To do so, we look to lower-bound $\Delta\mathcal{C}_{\ell_1}$ over the infimum of $r(x)$. Since $\Delta\mathcal{C}_{\ell_1}$ is a (strictly) convex function of $r(x)$, if we can select the parameters $\alpha$ and $\beta$ to ensure $r^*(x) > 0 \Leftrightarrow r_0(x) > 0$, where $r_0$ is the Bayes solution for the surrogate rejection loss, then this infimum occurs at $r(x) = 0$. This is illustrated in Figure 1. Proposition 4 states that this can be arranged if $\alpha$ and $\beta$ are related by $\frac{2\beta c}{\alpha} = \bar{I}_c$. In view of this proposition, we will adopt the assumption $\frac{2\beta c}{\alpha} = \bar{I}_c$ and analyze $\Delta\mathcal{C}_{\ell_1}(0, x)$. Note that the equivalence proven in the proof holds if and only if this equality holds.

**Theorem 5.** *Let $\alpha, \beta > 0$ be such that $\frac{2\beta c}{\alpha} = \bar{I}_c$, where $\bar{I}_c = ce^{\frac{\alpha}{2}} + (1 - c)e^{-\frac{\alpha}{2}}$. Then, the following inequality holds for any $r \in \mathcal{R}_{\mathrm{all}}$:*

$$R_{\ell_2}(r) - R_{\ell_2}^* \leq \frac{2}{e^{\frac{\alpha}{2}} - e^{-\frac{\alpha}{2}}} \sqrt{\frac{(c + \bar{I}_c)\bar{I}_c}{c}\big(R_{\ell_1}(r) - R_{\ell_1}^*\big)}.$$

See Appendix A.3 for the proof. The theorem shows that if the excess surrogate loss $\big(R_{\ell_1}(r) - R_{\ell_1}^*\big)$ is reduced to $\epsilon$, then the excess rejection loss $\big(R_{\ell_2}(r) - R_{\ell_2}^*\big)$ is bounded by $O(\sqrt{\epsilon})$. This provides a strong guarantee for the surrogate rejection loss function proposed when the condition $\frac{2\beta c}{\alpha} = \bar{I}_c$

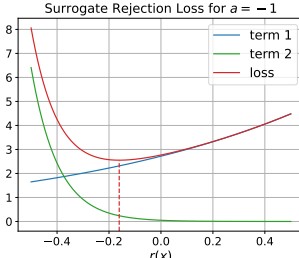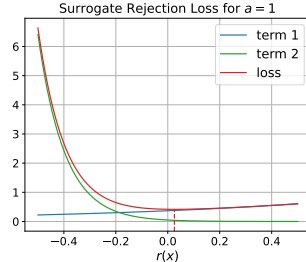

Figure 2: Surrogate rejection loss as a function of $r(x)$ when $c = 0.05, \alpha = 2$, and $\beta$ following $\frac{2\beta c}{\alpha} = \bar{I}_c$, with both terms in the sum. **Left:** negatively-labelled points (-1). **Right:** positively-labelled points (+1).

holds. Similar results can be derived for other families of functions $\mathcal{R}$, such as that of linear functions or neural networks with one hidden-layer as in (Awasthi et al., 2022a) (see Appendix D). This gives a *principled* method for defining the relation between $\alpha$ and $\beta$. The value of the other parameter, say $\alpha$, can be set arbitrarily or via a hyper-parameter search.

### 5.5 VISUALIZATION OF SURROGATE LOSS

In Figure 2, we plot our surrogate loss as a function of $r(x)$ on $(-0.5, +0.5)$, and arbitrarily choose $c = 0.05$ and $\alpha = 2$ with $\beta$ following the relationship defined in $\frac{2\beta c}{\alpha} = \bar{I}_c$. We include the plot for both negatively-annotated points ($a = -1$) and positively-annotated points ($a = +1$). The first term is always increasing, and the second is always decreasing.

We observe the following property: for negatively-annotated points, the minimum is attained at $r(x) < 0$, and for positively-annotated points, the minimum is attained at $r(x) > 0$. The following is a key insight from our $\mathcal{H}$-consistency bound: this property holds for any $c$, and any $\alpha$ and $\beta$ satisfying $\frac{2\beta c}{\alpha} = \bar{I}_c$, as the signs of $r^*(x)$ and $r_0(x)$ match. This relationship thus ensures that the minimums of both plots are in the proper regions.

## 6 DECONTEXTUALIZATION TASK

We choose the NLP task of decontextualization as a primary case study. This is a challenging task because only a modest amount of annotated data is available and because each input typically admits a large number of correct labels. We give a description of the task and our annotation procedure.

### 6.1 DEFINITION

Decontextualization is the task of editing a sentence within a passage so that it can be interpretable out of context (Choi et al., 2021). Specifically, given a sentence-context pair $(s, c)$, a sentence $s'$ is a valid decontextualization of $s$ if: (1) the sentence $s'$ is interpretable in the empty context; and (2) the truth-conditional meaning of $s'$ in the empty context is the same as the truth-conditional meaning of $s$ in context $c$. We refer readers to (Choi et al., 2021) for a full description.

### 6.2 ANNOTATION

For our experiments, we labeled 2,000 decontextualizations of a fixed MT5 XXL model (Xue et al., 2020) ourselves, fine-tuned on the decontextualization task. The training data for the original decontextualization task is a sample from the English portion of the Wikipedia corpus (Choi et al., 2021). The input is formed by concatenating the title and subtitle of the relevant page with '[HEAD]', and then appending the relevant paragraph after '[SEP]', see Figure 3. Our 2,000 annotated decontextualizations are originally from a random sample of English Wikipedia. Examples that the model labelled as 'impossible' or 'unnecessary' were not considered. We observe that annotating for this task is difficult: some take several minutes to evaluate.

When labeling or evaluating the validity of a decontextualization, we consider the correctness of the edits: the added information must be correct, and the deletions cannot change the meaning of

> Goldie Taylor [HEAD] Career ; Corporate [SEP] Taylor has worked for the Sara Lee Corporation as director of global communications and public affairs. **Goldie Taylor has served as executive consultant to NBC News and CNN Worldwide.**

Figure 3: Decontextualization labeled as 'title'.

the sentence. Sometimes, however, this is impossible to discern without using information from the title or subtitle. We thus labeled the outputs as 'yes', 'no' or 'title'. We present a 'title' example in Figure 3. The decontextualization request is for the bolded sentence, and 'Goldie' is then inserted. However, 'Goldie' only appears in the title. While it is probable that 'Taylor' refers to 'Goldie Taylor', we must rely on the information from the title. It is however also possible that 'Taylor' refers to a family member of 'Goldie Taylor' and that the paragraph is entirely unrelated to the title.

In our case, since 'title' examples are likely to be factual (while unsupported by the context provided to the model), we evaluate experimentally by including them with 'yes'. In other setting such as novels, 'title' examples are less likely to be factual, as paragraphs deep within a chapter have little connection to their title.

## 7 EXPERIMENTAL EVALUATION

In this section, we report results for the described learning methods.

### 7.1 DATASET

We randomly split our 2,000 annotation examples into 1,500 train/500 validation examples and perform 4-fold cross-validation. 1,019 (50.95%) of the annotations are 'yes', 761 (38.05%) are 'title', and the remaining 220 (11.00%) are 'no.' As already mentioned, we consider the 'title' examples as 'yes', so we have about 89% positive examples. The *decontextualization rejection task* is constructed as follows: we concatenate the input and output of the decontextualization model with the token '[OUT]' to form the input. The target consists of just one token, 'yes' or 'no.'

### 7.2 METHODS

**Maxprob**: We use the score that the fixed MT5 XXL predictor assigned to its own output sequence. The best threshold for some precision level is determined on the training data, and then evaluated on the validation data for both coverage and precision. A new threshold is determined for each split.

**Cross-entropy loss**: We further fine-tune a T5X 1.1 XXL decontextualization model (Roberts et al., 2022), limited to one output token, on the decontextualization rejection task, and use as the score the value of the $\text{logits}_{yes}$. The standard cross-entropy loss function is used, and a threshold is similarly fitted on half of the validation data and evaluated on the other half. We perform a hyper-parameter search over $\{1e-4, 1e-3, 1e-2\}$ for the learning rate, and $\{0, 0.05, \ldots, 0.2\}$ for the dropout rate.

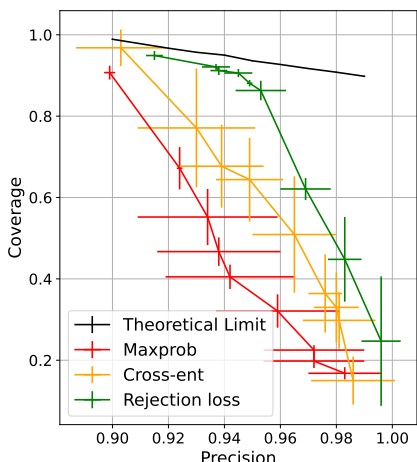

Figure 4: Precision vs. coverage on decontextualization. Standard deviations for both precision and coverage are from the 4 cross-validation splits.

**Surrogate loss**: In our formulation, we have a rejector $r: \mathcal{X} \to \mathbb{R}$. Thus, to convert the output of a T5X model to a real number for $r(x)$, we simply use $r(x) = \text{softmax}(\text{logits}(x))_{yes} - 0.5$. We further fine-tune the same model, but with our surrogate loss function: $e^{\frac{\alpha}{2}[r(x)-a]} + ce^{-\beta r(x)}$. In Figure 4, each point corresponds to a model trained with a value of $c \in \{0.02, 0.03, 0.04, 0.05, 0.07, 0.1, 0.15\}$. For the two most extreme points on the curve, we fit a threshold slightly different from 0 on the models for $c = 0.02$ and $c = 0.15$. We set $\alpha$ to 4, and do not perform a hyper-parameter search.

**Theoretical limit**: The theoretical limit of coverage can be defined as $b/p$, where $b$ is the fraction of positively labeled points and $p$ is the desired precision level. The precision is obtained exactly, and standard deviations for coverage are a result of the slightly different class imbalances in the cross-validation splits.

## 7.3 DISCUSSION

The performance of the four methods is reported in Figure 4. The details of the surrogate loss are shown in Table 1 and the details for various baselines are presented in Appendix B.1 (see Table 2).

We observe that our rejection loss clearly outperforms the baselines considered, and for the lower half of the precision levels, *closely follows the theoretical limit*. We provide, at $c = 0.07$ for example, about a halving of the error rate (11.00% to 5.1%) while maintaining a broad 90.6% coverage. The theoretical limit for 5% error is only slightly higher at 93.6% coverage, and

Table 1: Precision vs. Coverage for surrogate rejection loss on decontextualization.

| c | Precision | Coverage |
|---|---|---|
| 0.15 | $0.915 \pm 0.003$ | $0.949 \pm 0.011$ |
| 0.15 | $0.937 \pm 0.005$ | $0.921 \pm 0.007$ |
| 0.10 | $0.938 \pm 0.003$ | $0.912 \pm 0.011$ |
| 0.07 | $0.945 \pm 0.005$ | $0.906 \pm 0.010$ |
| 0.05 | $0.949 \pm 0.001$ | $0.881 \pm 0.008$ |
| 0.04 | $0.953 \pm 0.009$ | $0.863 \pm 0.024$ |
| 0.03 | $0.969 \pm 0.009$ | $0.621 \pm 0.027$ |
| 0.02 | $0.983 \pm 0.006$ | $0.448 \pm 0.104$ |
| 0.02 | $0.996 \pm 0.007$ | $0.247 \pm 0.159$ |

the closest baseline, the cross-entropy loss, only provides ~64.4% coverage at ~5.1% error. Another important observation is the stability of this result. Not only does the surrogate loss perform much better on average, but the standard deviations are also significantly smaller.

## 7.4 IMAGE CLASSIFICATION

While our focus is on LLM predictors and multiple correct labels, we provide additional empirical evaluation on two simpler image classification datasets: Fashion-MNIST (Xiao et al., 2017) and KMNIST (Clanuwat et al., 2018). These are standard 10-class classification tasks, both with $60,000$ training samples and $10,000$ test samples. Half of the training data is used for training a predictor, and the other half for training a rejector. We use a 5-layer fully-connected neural network. While this is a much easier task than decontextualiztion, and the model class is far less complex, we still observe an improvement when minimizing our surrogate rejection loss (see Figure 5). Due to fewer computational constraints, we include a cost-sensitive baseline, which uses the cross-entropy loss but reweights positive class by $c/(1-c)$ for varying $c$. The full details are deferred to Appendix B.2.

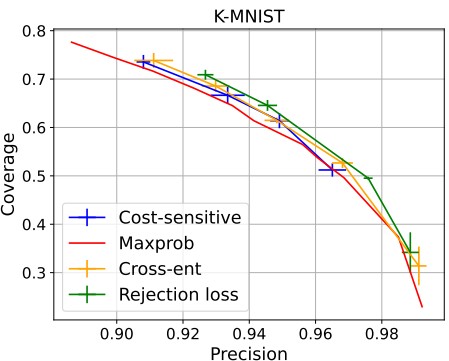

Figure 5: Precision vs. coverage on KM-NIST. Standard deviations for precision and coverage are from 4 training runs.

## 8 CONCLUSION

We presented a theoretically-justified approach to classification with a reject option for applications where the predictor remains fixed. Our main contributions include the following: (1) a new formulation of the problem of learning a rejector with fixed predictor (for cases where there may be many correct labels); (2) introduction of a new surrogate loss function for our scenario, with the proof of a strong $\mathcal{H}$-consistency bound guarantee; (3) definition of the notion of correctness for decontextualization, and its use to provide a dataset of 2,000 manually-labeled decontextualizations; (4) experimental results demonstrating a 10-25% improvement over baselines in coverage at various precision levels on decontextualization.

We observe that our algorithm can be used in other settings where a binary label indicating correctness of a prediction is available. Annotation in NLP can of course be expensive, and this limits the breadth of our experimental evaluation. We encourage the use of our algorithm in difficult rejection tasks with a large output space and a large number of correct labels. In particular, our algorithm can be used for abstention with large language models, in a variety of contexts such as text generation.

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

## A  $\mathcal{H}$-CONSISTENCY BOUND PROOF

### A.1  CALIBRATION GAP FOR REJECTION LOSS

The following gives the expression of the calibration gap $\Delta \mathcal{C}_{\ell_2}$.

**Lemma 2.** *The Bayes solution $r^*$ for the rejection loss can be expressed for all $x \in \mathcal{X}$ by $r^*(x) = \eta(x) - (1 - c)$. The calibration gap for the rejection loss is given for any $r \in \mathcal{R}_{\text{all}}$ and $x \in \mathcal{X}$ by*

$$\Delta \mathcal{C}_{\ell_2}(r, x) = |\eta(x) - (1 - c)| \mathbb{I}_{r(x)r^*(x) \le 0}.$$

*Proof.* For any $r \in \mathcal{R}_{\text{all}}$ and $x \in \mathcal{X}$, we can write

$$\mathcal{C}_{\ell_2}(r, x) = \eta(x)\ell_2(r, x, +1)$$
$$+ [1 - \eta(x)]\ell_2(r, x, -1)$$
$$= \eta(x)\left[\mathbb{I}_{+1=-1}\mathbb{I}_{r(x)>0} + c\mathbb{I}_{r(x)\le 0}\right]$$
$$+ [1 - \eta(x)]\left[\mathbb{I}_{-1=-1}\mathbb{I}_{r(x)>0} + c\mathbb{I}_{r(x)\le 0}\right]$$
$$= c\mathbb{I}_{r(x)\le 0} + [1 - \eta(x)]\mathbb{I}_{r(x)>0}.$$

For the optimal $\mathcal{C}_{\ell_2}^*$, we would always pick the lower of $c$ or $1 - \eta(x)$, which gives: $\mathcal{C}_{\ell_2}^*(x) = \min\{c, 1 - \eta(x)\}$. The corresponding Bayes solution $r^*$ can be defined by $r^*(x) = \eta(x) - (1 - c)$. Thus, the calibration gap is given by

$$\Delta \mathcal{C}_{\ell_2}(r, x) = c\mathbb{I}_{r(x)\le 0} + [1 - \eta(x)]\mathbb{I}_{r(x)>0}$$
$$- \min\{c, 1 - \eta(x)\}.$$

If $r(x)$ correctly chooses the lower of the two, we have $r(x)r^*(x) > 0$ and then $\Delta \mathcal{C}_{\ell_2} = 0$. Otherwise,

$$\Delta \mathcal{C}_{\ell_2}(r, x) = \begin{cases} c - (1 - \eta(x)) & \text{if } r(x) \le 0 \\ (1 - \eta(x)) - c & \text{otherwise} \end{cases}.$$

Thus, for all $x \in \mathcal{X}$, we have $\Delta \mathcal{C}_{\ell_2}(r, x) = |\eta(x) - (1 - c)| \mathbb{I}_{r(x)r^*(x)\le 0}$. This completes the proof. $\square$

### A.2  CALIBRATION GAP FOR SURROGATE LOSS

Here, we analyze the calibration gap for the surrogate loss.

**Lemma 3.** *Let $I_\eta(x)$ be defined by $I_\eta(x) = \eta(x)e^{-\frac{\alpha}{2}} + (1 - \eta(x))e^{\frac{\alpha}{2}}$ and define $\gamma$ by $\gamma = \frac{\alpha}{\alpha + 2\beta}$. Then, the calibration gap for the surrogate loss is given for any $r \in \mathcal{R}_{\text{all}}$ and $x \in \mathcal{X}$ by*

$$\Delta \mathcal{C}_{\ell_1}(r, x) = e^{\frac{\alpha}{2}r(x)} I_\eta(x) + ce^{-\beta r(x)} - \frac{1}{1 - \gamma}\left(\frac{2\beta c}{\alpha}\right)^\gamma I_\eta(x)^{1-\gamma}.$$

*Proof.* By definition, the calibration function for $\ell_1$ can be expressed for all $x \in \mathcal{X}$ by

$$\mathcal{C}_{\ell_1}(r, x) = \eta(x)\ell_1(r, x, +1)$$
$$+ [1 - \eta(x)]\ell_1(r, x, -1)$$
$$= \eta(x)\left[e^{\frac{\alpha}{2}[r(x)-1]} + ce^{-\beta r(x)}\right]$$
$$+ [1 - \eta(x)]\left[e^{\frac{\alpha}{2}[r(x)+1]} + ce^{-\beta r(x)}\right]$$
$$= e^{\frac{\alpha}{2}r(x)} I_\eta(x) + ce^{-\beta r(x)}.$$

Since the exponential function is convex, $\Delta \mathcal{C}_{\ell_1}(r, x)$ is a convex function of $r(x)$. Thus, for $r \in \mathcal{R}_{\text{all}}$, we obtain the minimum $r_0(x)$ by differentiating with respect to $r(x)$ and setting to 0:

$$\frac{\alpha}{2}e^{\frac{\alpha}{2}r(x)} I_\eta(x) - \beta ce^{-\beta r(x)} = 0$$

$$\Leftrightarrow r_0(x) = \log\left[\left(\frac{2\beta c}{\alpha I_\eta(x)}\right)^{\frac{2}{2\beta + \alpha}}\right].$$

Plugging in this expression in $\mathcal{C}_{\ell_1}$ gives the corresponding minimal calibration $\mathcal{C}^*_{\ell_1}(x)$: $\mathcal{C}^*_{\ell_1}(x) = \left[\left(\frac{2\beta c}{\alpha}\right)^\gamma\right] I_\eta(x)^{1-\gamma}\left(\frac{1}{1-\gamma}\right)$. This completes the proof. $\qquad\square$

### A.3 $\mathcal{H}$-CONSISTENCY BOUND

In this section, we prove our main result. The following will provide a key tool to derive our result.

**Proposition 4.** *Assume that there exists a convex function* $\Psi\colon \mathbb{R}_+ \to \mathbb{R}$ *with* $\Psi(0) = 0$ *such that the following holds for all* $r \in \mathcal{R}_{\mathrm{all}}$ *and* $x \in \mathcal{X}$: $\Psi\big(|\eta(x) - (1-c)|\,\mathbb{I}_{r(x)r^*(x)\leq 0}\big) \leq \Delta\mathcal{C}_{\ell_1}(0,x)$. *Let* $\bar{I}_c$ *be defined by* $\bar{I}_c = ce^{\frac{\alpha}{2}} + (1-c)e^{-\frac{\alpha}{2}}$ *and assume that* $\frac{2\beta c}{\alpha} = \bar{I}_c$. *Then, for any* $r \in \mathcal{R}_{\mathrm{all}}$:

$$\Psi\big(R_{\ell_2}(r) - R^*_{\ell_2}\big) \leq R_{\ell_1}(r) - R^*_{\ell_1}. \tag{3}$$

*Proof.* We will show that the following holds: $\inf_{r(x)r^*(x)\leq 0} \Delta\mathcal{C}_{\ell_1}(r,x) = \Delta\mathcal{C}_{\ell_1}(0,x)$. The result then follows by Theorem 1 and Lemma 2. Since we have $\frac{2\beta c}{\alpha} = \bar{I}_c$, the following equivalence holds:

$$r_0(x) > 0 \Leftrightarrow \frac{2\beta c}{\alpha I_\eta(x)} > 1$$

$$\Leftrightarrow I_\eta(x) < \bar{I}_c$$

$$\Leftrightarrow \eta(x) > \frac{e^{\frac{\alpha}{2}} - \bar{I}_c}{e^{\frac{\alpha}{2}} - e^{-\frac{\alpha}{2}}}$$

$$\Leftrightarrow \eta(x) > \frac{(1-c)e^{\frac{\alpha}{2}} - (1-c)e^{-\frac{\alpha}{2}}}{e^{\frac{\alpha}{2}} - e^{-\frac{\alpha}{2}}}$$

$$\Leftrightarrow r^*(x) > 0.$$

This implies $\inf_{r(x)r^*(x)\leq 0} \mathcal{C}_{\ell_1}(r,x) = \inf_{r(x)r_0(x)\leq 0} \mathcal{C}_{\ell_1}(r,x)$. Now, since $r_0(x)$ is the unique minimizer of the strictly convex function $\mathcal{C}_{\ell_1}(r,x)$ of $r(x)$, then, as a function of $r(x)$, $\mathcal{C}_{\ell_1}(r,x)$ is decreasing from $-\infty$ to $r_0(x)$ and increasing from there to $+\infty$. Thus, if $r_0(x) > 0$, the infimum of $\mathcal{C}_{\ell_1}(r,x)$ over $r(x) \leq 0$ is reached for $r(x) = 0$. Similarly, if $r_0(x) < 0$, the infimum of $\mathcal{C}_{\ell_1}(r,x)$ over $r(x) \geq 0$ is reached for $r(x) = 0$. This shows that $\inf_{r(x)r_0(x)\leq 0} \mathcal{C}_{\ell_1}(r,x) = \mathcal{C}_{\ell_1}(0,x)$, and completes the proof. $\qquad\square$

The proof of our main result makes use of the following identity, which is a refinement of Bernoulli's inequality. The result could be of independent interest in other contexts, we give a concise proof below.

**Lemma 6** (Bernoulli-type inequality). *The following identity holds for all* $x, r \in (0,1)$,

$$(1+x)^r \leq 1 + rx + \frac{r(r-1)x^2}{4}.$$

*Proof.* Let $f_r(x) = (1+x)^r - \left(1 + rx + \frac{r(r-1)x^2}{4}\right)$. We will show that $f_r(x) \leq 0$ for all $x, r \in (0,1)$. We have $f'_r(x) = r(1+x)^{r-1} - \left(r + \frac{r(r-1)x}{2}\right)$, and $f'_r(0) = 0$. To see that $f'_r(1) \leq 0$, observe $r2^{r-1} - \left(r + \frac{r(r-1)}{2}\right) \leq 0 \Leftrightarrow 2^{r-1} - \frac{(r-1)}{2} \leq 1$. The left-hand side of the last inequality is a convex function of $r$, and equal to 1 when $r = 0$ or $r = 1$. Thus, the left-hand side is less than or equal 1 for $r \in (0,1)$, giving $f'_r(1) \leq 0$. Since $f'_r(x)$ is a convex function of $x$, with $f'_r(0) \leq 0$ and $f'_r(1) \leq 0$, then $f'_r(x) \leq 0$ for all $x \in (0,1)$, which shows $f_r$ is decreasing. Then, since $f_r(0) = 0$, $f_r(x) \leq 0$ for all $x, r \in (0,1)$. $\qquad\square$

The following is our main result; it relates the surrogate excess error to that of the rejection loss.

**Theorem 5.** *Let* $\alpha, \beta > 0$ *be such that* $\frac{2\beta c}{\alpha} = \bar{I}_c$, *where* $\bar{I}_c = ce^{\frac{\alpha}{2}} + (1-c)e^{-\frac{\alpha}{2}}$. *Then, the following inequality holds for any* $r \in \mathcal{R}_{\mathrm{all}}$:

$$R_{\ell_2}(r) - R^*_{\ell_2} \leq \frac{2}{e^{\frac{\alpha}{2}} - e^{-\frac{\alpha}{2}}}\sqrt{\frac{(c + \bar{I}_c)\bar{I}_c}{c}\big(R_{\ell_1}(r) - R^*_{\ell_1}\big)}.$$

*Proof.* Using the expression of $\Delta\mathcal{C}_{\ell_1}$ given by Lemma 3, we can write

$$\Delta\mathcal{C}_{\ell_1}(0,x) = I_\eta(x) + c - \frac{1}{1-\gamma}\left(\frac{2\beta c}{\alpha}\right)^\gamma I_\eta(x)^{1-\gamma}$$

$$= I_\eta(x) + c - \left(\bar{I}_c + c\right)\left(\frac{I_\eta(x)}{\bar{I}_c}\right)^{1-\gamma}.$$

We can express this formula in terms of $u(x) = \eta(x) - (1-c)$, using $I_\eta(x) = J_u(x) + \bar{I}_c$, with $J_u(x) = \left[e^{-\frac{\alpha}{2}} - e^{\frac{\alpha}{2}}\right]u(x)$:

$$\Delta\mathcal{C}_{\ell_1}(0,x)$$

$$= J_u(x) + \bar{I}_c + c - \left(\bar{I}_c + c\right)\left[1 + \frac{J_u(x)}{\bar{I}_c}\right]^{1-\gamma}$$

$$\geq \frac{\bar{I}_c}{c + \bar{I}_c}\frac{c}{c + \bar{I}_c}\frac{c + \bar{I}_c}{4}\left[\frac{J_u(x)}{\bar{I}_c}\right]^2$$

$$= \frac{1}{4}\frac{c\bar{I}_c}{c + \bar{I}_c}\left[\frac{J_u(x)}{\bar{I}_c}\right]^2.$$

where we used Lemma 6. The function $\Psi(u)$ defined by this expression verifies the condition of Proposition 4 and therefore we have $\Psi\left(R_{\ell_2}(h) - R_{\ell_2}^*\right) \leq R_{\ell_1}(h) - R_{\ell_1}^*$. An explicit upper-bound on $R_{\ell_2}(h) - R_{\ell_2}^*$ can be written in terms of $\Psi^{-1}$: $R_{\ell_2}(h) - R_{\ell_2}^* \leq \Psi^{-1}\left(R_{\ell_1}(h) - R_{\ell_1}^*\right)$. To derive the expression of $\Psi^{-1}$, we write $z = \Psi(u)$, that is:

$$4\frac{c + \bar{I}_c}{c\bar{I}_c}z = \left[\frac{u(x)}{\bar{I}_c}\right]^2\left[e^{\frac{\alpha}{2}} - e^{-\frac{\alpha}{2}}\right]^2$$

$$\Leftrightarrow |u| = \frac{2}{e^{\frac{\alpha}{2}} - e^{-\frac{\alpha}{2}}}\sqrt{\frac{(c + \bar{I}_c)\bar{I}_c}{c}z}.$$

Thus, we have, for all $r \in \mathcal{R}_{\text{all}}$, $R_{\ell_2}(r) - R_{\ell_2}^* \leq \frac{2}{e^{\frac{\alpha}{2}} - e^{-\frac{\alpha}{2}}}\sqrt{\frac{(c+\bar{I}_c)\bar{I}_c}{c}\left(R_{\ell_1}(r) - R_{\ell_1}^*\right)}.$ $\qquad\square$

# B  EXPERIMENTAL DETAILS

## B.1  DECONTEXTUALIZATION

In this section, we report the detailed results for our experiments on the decontextualization task. Table 2 presents the mean and standard deviation of the precision and coverage of various baselines over 4 cross-validation splits and Table 1 in Section 7.3 provides detailed results of the surrogate loss.

Table 2: Precision vs. Coverage for various baselines on decontextualization, with theoretical limit.

| Target precision | MAXPROB precision | coverage | CROSS-ENTROPY precision | coverage | THEORETICAL LIMIT precision | coverage |
|---|---|---|---|---|---|---|
| 0.90 | $0.899 \pm 0.002$ | $0.907 \pm 0.017$ | $0.903 \pm 0.016$ | $0.968 \pm 0.045$ | 0.90 | $0.989 \pm 0.001$ |
| 0.92 | $0.924 \pm 0.001$ | $0.672 \pm 0.052$ | $0.930 \pm 0.021$ | $0.771 \pm 0.146$ | 0.92 | $0.967 \pm 0.001$ |
| 0.93 | $0.934 \pm 0.025$ | $0.552 \pm 0.069$ | $0.939 \pm 0.015$ | $0.677 \pm 0.102$ | 0.93 | $0.957 \pm 0.001$ |
| 0.94 | $0.938 \pm 0.022$ | $0.467 \pm 0.035$ | $0.949 \pm 0.012$ | $0.644 \pm 0.103$ | 0.94 | $0.950 \pm 0.001$ |
| 0.95 | $0.942 \pm 0.023$ | $0.405 \pm 0.030$ | $0.965 \pm 0.015$ | $0.509 \pm 0.143$ | 0.95 | $0.936 \pm 0.001$ |
| 0.96 | $0.959 \pm 0.022$ | $0.321 \pm 0.041$ | $0.976 \pm 0.006$ | $0.364 \pm 0.096$ | 0.96 | $0.927 \pm 0.001$ |
| 0.97 | $0.972 \pm 0.018$ | $0.225 \pm 0.012$ | $0.980 \pm 0.008$ | $0.330 \pm 0.086$ | 0.97 | $0.917 \pm 0.001$ |
| 0.98 | $0.972 \pm 0.018$ | $0.198 \pm 0.017$ | $0.981 \pm 0.013$ | $0.298 \pm 0.069$ | 0.98 | $0.908 \pm 0.001$ |
| 0.99 | $0.983 \pm 0.013$ | $0.168 \pm 0.015$ | $0.986 \pm 0.015$ | $0.150 \pm 0.059$ | 0.99 | $0.898 \pm 0.001$ |

## B.2  IMAGE CLASSIFICATION

In this section, we provide details of our experiments on Fashion-MNIST, a fashion image dataset, and KMNIST, a cursive Japanese letter dataset. Both are perfectly balanced between their 10 classes. In both cases, we use a 5-layer fully-connected neural network to train a predictor with half of the training data. The remaining half is reserved for the rejector. Training the rejector is a binary classification task: for pairs $(x, y)$ occuring in the usual dataset, we construct another dataset $((x, f_p(x)), \mathbb{I}_{f(x)=y})$, where $f$ is the predictor and $f_p(x)$ is the probability that $f$ assigns to its prediction on $x$. In our experiments, we observe that it is important to append $f_p(x)$ as a feature to $x$. Note that constructing this binary classification dataset does not require manual annotation. For Fashion-MNIST, our predictor is trained to $85.3\%$ accuracy on its test set, and for KMNIST, our predictor is trained to $79.1\%$ accuracy on its test set. While it is possible to improve the performance of these predictors, this is not our focus. We are focused on a rejection task given some fixed predictor.

Next, we detail the methods for rejection.

**Maxprob.** Similar to the decontextualization experiment, we fit thresholds on the scores assigned by the predictor. Since this method is deterministic (and the error bars here are over rejector training runs), there are no error bars to report.

**Cross-entropy loss.** We train another 5-layer neural network on the constructed binary classification dataset using the cross-entropy loss. Similar to the decontextualization experiment, thresholds are fitted on the scores of this neural netowrk.

**Rejection loss.** We train a second 5-layer neural network on the constructed binary classification dataset using our proposed surrogate rejection loss. For Fashion-MNIST, $c$ is varied in $\{0.05, 0.1, 0.2, 0.3, 0.5\}$. For KMNIST, $c$ is varied in $\{0.025, 0.05, 0.1, 0.15\}$. Each point on the plot represents a model trained with a different value of $c$. We set $\alpha$ in the surrogate rejection loss function to $3.5$.

**Cost-sensitive loss.** We train a third 5-layer neural network on the constructed binary classification dataset using the cross-entropy loss, but with the positive class reweighted by $c/(1-c)$. For Fashion-MNIST, $c$ is varied in $\{0.05, 0.1, 0.2, 0.3, 0.5\}$. For KMNIST, $c$ is varied in $\{0.03, 0.05, 0.1, 0.2\}$. Each point on the plot represents a model trained with a different value of $c$.

For all methods, we use the Adam optimizer (Kingma and Ba, 2014), and tune the learning rate in $[1e-4, 1e-7]$ and number of epochs in $[20, 100]$.

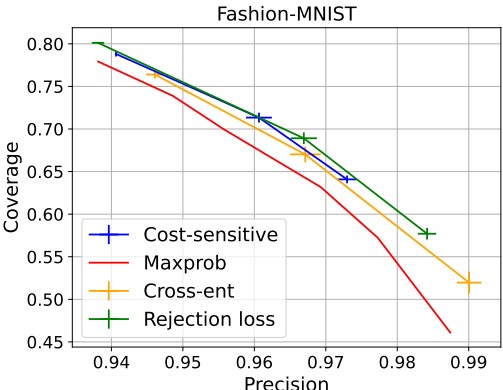

Figure 6: Precision vs. Coverage on Fashion-MNIST. Standard deviations for both precision and coverage are from 4 different training runs.

The precision vs. coverage graph for KMNIST is reported in Figure 5 in Section 7.4 and the precision vs. coverage graph for Fashion-MNIST is reported in Figure 6. We do not plot the theoretical limit since no method is near it in this setting. In both cases, we generally observe that the rejection loss lies above the baselines. It is likely that the predictor in this setting is much better calibrated than a large language model, and thus Maxprob is a much stronger baseline with not as much room for improvement as on decontextualization. We also note that it may also be possible to improve the performance of our method by tuning $\alpha$.

## C  COMPARISON WITH COST-SENSITIVE CLASSIFICATION

It is worth pointing out that minimizing the induced rejection loss is equivalent to minimizing a cost-sensitive classification loss (Elkan, 2001; Steinwart, 2007; Scott, 2012; Charoenphakdee et al., 2021), since by using the decomposition $\mathbb{I}_{a=-1}\,\mathbb{I}_{r(x)>0} = (1-c)\,\mathbb{I}_{a=-1}\,\mathbb{I}_{r(x)>0} + c\,\mathbb{I}_{a=-1}\,\mathbb{I}_{r(x)>0}$ and $c\,\mathbb{I}_{r(x)\leq0} = c\,\mathbb{I}_{a=+1}\,\mathbb{I}_{r(x)\leq0} + c\,\mathbb{I}_{a=-1}\,\mathbb{I}_{r(x)\leq0}$, the loss (2) can be rewritten as

$$\ell(r,x,a) = \mathbb{I}_{a=-1}\,\mathbb{I}_{r(x)>0} + c\,\mathbb{I}_{r(x)\leq0}$$

$$= (1-c)\,\mathbb{I}_{a=-1}\,\mathbb{I}_{r(x)>0} + c\,\mathbb{I}_{a=-1}\,\mathbb{I}_{r(x)>0} + c\,\mathbb{I}_{a=+1}\,\mathbb{I}_{r(x)\leq0} + c\,\mathbb{I}_{a=-1}\,\mathbb{I}_{r(x)\leq0}$$

$$= (1-c)\,\mathbb{I}_{a=-1}\,\mathbb{I}_{r(x)>0} + c\,\mathbb{I}_{a=+1}\,\mathbb{I}_{r(x)\leq0} + c\,\mathbb{I}_{a=-1},$$

where in the last step we use the fact that $c\,\mathbb{I}_{a=-1}\,\mathbb{I}_{r(x)\leq0} + c\,\mathbb{I}_{a=-1}\,\mathbb{I}_{r(x)>0} = c\,\mathbb{I}_{a=-1}$. In light of this expression, since the last term $c\,\mathbb{I}_{a=-1}$ does not depend on $r$, if $x \mapsto \phi(-x)$ is a convex function upper-bounding $\mathbb{I}_{x\leq0}$, then, $\ell_\phi$ defined as follows for any $r \in \mathcal{R}$ and $(x,a) \in \mathcal{X} \times \{-1,+1\}$, is a natural surrogate loss for $\ell$:

$$\ell_\phi(r,x,a) = (1-c)\,\mathbb{I}_{a=-1}\,\phi(r(x)) + c\,\mathbb{I}_{a=+1}\,\phi(-r(x)).$$

We will refer to $\ell_\phi$ as cost-sensitive surrogate losses for the induced rejection loss. However, this cost-sensitive approach suffers from several issues: (i) There is a lack of any $\mathcal{H}$-consistency bound guarantees for cost-sensitive surrogate losses with respect to the induced rejection loss. Conversely, our theoretical analysis can potentially extend to an $\mathcal{H}$-consistent surrogate loss function for cost-sensitive classification. This would provide a theoretically justified algorithm for that context. Our novel contribution lies in introducing a loss function for the induced rejection loss backed by strong $\mathcal{H}$-consistency bounds; (ii) It has been shown in (Cao et al., 2022) that the cost-sensitive approach (Charoenphakdee et al., 2021) can not produce the state-of-the-art performance in the learning with rejection framework, which motivates us to propose a new theoretically guaranteed surrogate loss in our rejection scenario; (iii) As shown in (Charoenphakdee et al., 2021), the cost-sensitive approach equivalently solves $n$ one-versus-all binary classification problems, where $n$ is the number of classes. Therefore, when the size of the sub-sample containing some of the classes is relatively small, the one-versus-all binary classification problem may face challenges due to insufficient data or increased risk of overfitting. This issue stands out for the decontextualization task, where the samples corresponding to $a = -1$ are much fewer than those corresponding to $a = +1$; (iv) Our empirical results on the benchmark datasets show that the cost-sensitive approach is inferior to our proposed surrogate loss function, which substantiate the effectiveness of our approach.

# D $\mathcal{H}$-CONSISTENCY BOUNDS BEYOND $\mathcal{H}_{\text{all}}$ AND PROOF

Here, we will show that our surrogate losses benefit from $\mathcal{R}$-consistency bounds with the hypothesis set $\mathcal{R}$ extending beyond the family of all measurable functions $\mathcal{R}_{\text{all}}$. Without loss of generality, we consider $\mathcal{X} = \{x \in \mathbb{R}^d \mid \|x\|_p \leq 1\}$. Let $p, q \in [1, +\infty]$ be conjugate numbers such that $\frac{1}{p} + \frac{1}{q} = 1$. We will consider *bounded* hypothesis sets $\mathcal{R}$, that is, there exists a function $\overline{r} \colon \mathcal{X} \to \mathbb{R}_+$ such that for all $r \in \mathcal{R}$ and $x \in \mathcal{X}$, $|r(x)| \leq \overline{r}(x)$, and all values in $[-\overline{r}(x), \overline{r}(x)]$ can be reached. As shown by Awasthi et al. (2022a), for the family of linear models $\mathcal{R}_{\text{lin}} = \{x \mapsto w \cdot x + b \mid \|w\|_q \leq W, |b| \leq B\}$ and one-hidden-layer ReLU networks $\mathcal{R}_{\text{NN}} = \{x \mapsto \sum_{j=1}^n u_j(w_j \cdot x + b_j)_+ \mid \|u\|_1 \leq \Lambda, \|w_j\|_q \leq W, |b_j| \leq B\}$, where $(\cdot)_+ = \max(\cdot, 0)$, we have $\overline{r}(x) = W\|x\|_p + B$ and $\overline{r}(x) = \Lambda W\|x\|_p + \Lambda B$ respectively.

## D.1 MAIN RESULT

In this section, we present our main result on $\mathcal{R}$-consistency bounds with bounded hypothesis sets $\mathcal{R}$ (Theorem 7), including $\mathcal{R}_{\text{lin}}$ and $\mathcal{R}_{\text{NN}}$ considered in (Awasthi et al., 2022a) as special cases (Corollary 8). The proofs are presented in Appendix D.4.

**Theorem 7.** *Assume that $\mathcal{R}$ is bounded with function $\overline{r} \colon \mathcal{X} \to \mathbb{R}$. Let $\alpha, \beta > 0$ be such that $\frac{2\beta c}{\alpha} = \overline{I}_c$, where $\overline{I}_c = c e^{\frac{\alpha}{2}} + (1-c)e^{-\frac{\alpha}{2}}$. Then, the following inequality holds for any $r \in \mathcal{R}$:*

$$R_{\ell_2}(r) - R_{\ell_2, \mathcal{R}}^* + \mathcal{M}_{\ell_2, \mathcal{R}} \leq \Gamma\big(R_{\ell_1}(r) - R_{\ell_1, \mathcal{R}}^* + \mathcal{M}_{\ell_1, \mathcal{R}}\big), \tag{4}$$

*where* $\Gamma(z) = \begin{cases} \dfrac{2}{e^{\frac{\alpha}{2}} - e^{-\frac{\alpha}{2}}}\sqrt{\dfrac{(c+\overline{I}_c)\overline{I}_c}{c}}z & 0 \leq z \leq \frac{1}{4}\dfrac{c\overline{I}_c}{c+\overline{I}_c}\Big[1 - e^{-\frac{\alpha(\overline{I}_c+c)\inf_{x\in\mathcal{X}}\overline{r}(x)}{2c}}\Big]^2 \\ \dfrac{4(c+\overline{I}_c)}{c\Big[1 - e^{-\frac{\alpha(\overline{I}_c+c)\inf_{x\in\mathcal{X}}\overline{r}(x)}{2c}}\Big]\Big[e^{\frac{\alpha}{2}} - e^{-\frac{\alpha}{2}}\Big]}z & otherwise. \end{cases}$

**Corollary 8.** *Let $\mathcal{R} = \mathcal{R}_{\text{lin}}$ or $\mathcal{R}_{\text{NN}}$. Let $\alpha, \beta > 0$ be such that $\frac{2\beta c}{\alpha} = \overline{I}_c$, where $\overline{I}_c = c e^{\frac{\alpha}{2}} + (1-c)e^{-\frac{\alpha}{2}}$. Then, the following inequality holds for any $r \in \mathcal{R}$:*

$$R_{\ell_2}(r) - R_{\ell_2, \mathcal{R}}^* + \mathcal{M}_{\ell_2, \mathcal{R}} \leq \Gamma\big(R_{\ell_1}(r) - R_{\ell_1, \mathcal{R}}^* + \mathcal{M}_{\ell_1, \mathcal{R}}\big), \tag{5}$$

*where* $\Gamma(z) = \begin{cases} \dfrac{2}{e^{\frac{\alpha}{2}} - e^{-\frac{\alpha}{2}}}\sqrt{\dfrac{(c+\overline{I}_c)\overline{I}_c}{c}}z & 0 \leq z \leq \frac{1}{4}\dfrac{c\overline{I}_c}{c+\overline{I}_c}\Big[1 - e^{-\frac{\alpha(\overline{I}_c+c)B}{2c}}\Big]^2 \\ \dfrac{4(c+\overline{I}_c)}{c\Big[1 - e^{-\frac{\alpha(\overline{I}_c+c)B}{2c}}\Big]\Big[e^{\frac{\alpha}{2}} - e^{-\frac{\alpha}{2}}\Big]}z & otherwise \end{cases}$ *and $B$ is replaced by* $\Lambda B$ *for $\mathcal{R} = \mathcal{R}_{\text{NN}}$.*

## D.2 CALIBRATION GAP FOR REJECTION LOSS

We first extend Lemma 2 to any hypothesis set $\mathcal{R}$ that is *regular for rejection*.

**Definition 9.** *We say that a hypothesis set $\mathcal{R}$ is* regular for rejection *if for all $x \in \mathcal{X}$, there exist $r_+, r_- \in \mathcal{R}$ such that $r_+(x) > 0$ and $r_+(x) \leq 0$.*

It is clear that all bounded hypothesis sets including $\mathcal{R}_{\text{lin}}$ and $\mathcal{R}_{\text{NN}}$ are regular for rejection. The following gives the expression of the calibration gap $\Delta\mathcal{C}_{\ell_2}$ for all hypothesis sets $\mathcal{R}$ that are regular for rejection. The proof is nearly identical to Lemma 2.

**Lemma 10.** *Assume that $\mathcal{R}$ is regular for rejection. The best-in-class solution $r^*$ for the rejection loss can be expressed for all $x \in \mathcal{X}$ by $r^*(x) = \eta(x) - (1-c)$. The calibration gap for the rejection loss is given for any $r \in \mathcal{R}$ and $x \in \mathcal{X}$ by*

$$\Delta\mathcal{C}_{\ell_2}(r, x) = |\eta(x) - (1-c)|\, \mathbb{I}_{r(x)r^*(x)\leq 0}.$$

*Proof.* For any $r \in \mathcal{R}$ and $x \in \mathcal{X}$, we can write

$$\mathcal{C}_{\ell_2}(r, x) = \eta(x)\ell_2(r, x, +1)$$
$$+ [1 - \eta(x)]\ell_2(r, x, -1)$$
$$= \eta(x)\left[\mathbb{I}_{+1=-1}\,\mathbb{I}_{r(x)>0} + c\,\mathbb{I}_{r(x)\leq 0}\right]$$
$$+ [1 - \eta(x)]\left[\mathbb{I}_{-1=-1}\,\mathbb{I}_{r(x)>0} + c\,\mathbb{I}_{r(x)\leq 0}\right]$$
$$= c\,\mathbb{I}_{r(x)\leq 0} + [1 - \eta(x)]\,\mathbb{I}_{r(x)>0}.$$

For the optimal $\mathcal{C}^*_{\ell_2,\mathcal{R}}$, since $\mathcal{R}$ is regular, we would always pick the lower of $c$ or $1 - \eta(x)$, which gives: $\mathcal{C}^*_{\ell_2,\mathcal{R}}(x) = \min\{c, 1 - \eta(x)\}$. The corresponding best-in-class solution $r^*$ can be defined by $r^*(x) = \eta(x) - (1 - c)$. Thus, the calibration gap is given by

$$\Delta\mathcal{C}_{\ell_2}(r, x) = c\,\mathbb{I}_{r(x)\leq 0} + [1 - \eta(x)]\,\mathbb{I}_{r(x)>0}$$
$$- \min\{c, 1 - \eta(x)\}.$$

If $r(x)$ correctly chooses the lower of the two, we have $r(x)r^*(x) > 0$ and then $\Delta\mathcal{C}_{\ell_2} = 0$. Otherwise,

$$\Delta\mathcal{C}_{\ell_2}(r, x) = \begin{cases} c - (1 - \eta(x)) & \text{if } r(x) \leq 0 \\ (1 - \eta(x)) - c & \text{otherwise} \end{cases}.$$

Thus, for all $x \in \mathcal{X}$, we have $\Delta\mathcal{C}_{\ell_2}(r, x) = |\eta(x) - (1 - c)|\,\mathbb{I}_{r(x)r^*(x)\leq 0}$. This completes the proof. $\square$

### D.3 CALIBRATION GAP FOR SURROGATE LOSS

Next, we extend Lemma 3 to bounded hypothesis sets $\mathcal{R}$. The following gives the expression of the calibration gap for the surrogate loss. The proof directly extends that of Lemma 3.

**Lemma 11.** *Assume that $\mathcal{R}$ is bounded with function $\bar{r}: \mathcal{X} \to \mathbb{R}$. Let $I_\eta(x) = \eta(x)e^{-\frac{\alpha}{2}} + (1-\eta(x))e^{\frac{\alpha}{2}}$, $r_0(x) = \log\left[\left(\frac{2\beta c}{\alpha I_\eta(x)}\right)^{\frac{2}{2\beta+\alpha}}\right]$ and $\gamma = \frac{\alpha}{\alpha+2\beta}$. Then, the calibration gap for the surrogate loss is given for any $r \in \mathcal{R}$ and $x \in \mathcal{X}$ by*

$$\Delta\mathcal{C}_{\ell_1}(r, x) = \begin{cases} e^{\frac{\alpha}{2}r(x)}I_\eta(x) + ce^{-\beta r(x)} - \frac{1}{1-\gamma}\left(\frac{2\beta c}{\alpha}\right)^\gamma I_\eta(x)^{1-\gamma} & -\bar{r}(x) \leq r_0(x) \leq \bar{r}(x) \\ e^{\frac{\alpha}{2}r(x)}I_\eta(x) + ce^{-\beta r(x)} - e^{\frac{\alpha}{2}\bar{r}(x)}I_\eta(x) - ce^{-\beta\bar{r}(x)} & r_0(x) > \bar{r}(x) \\ e^{\frac{\alpha}{2}r(x)}I_\eta(x) + ce^{-\beta r(x)} - e^{-\frac{\alpha}{2}\bar{r}(x)}I_\eta(x) - ce^{\beta\bar{r}(x)} & r_0(x) < -\bar{r}(x). \end{cases}$$

*Proof.* By definition, the calibration function for $\ell_1$ can be expressed for all $x \in \mathcal{X}$ by

$$\mathcal{C}_{\ell_1}(r, x) = \eta(x)\ell_1(r, x, +1) + [1 - \eta(x)]\ell_1(r, x, -1)$$
$$= \eta(x)\left[e^{\frac{\alpha}{2}[r(x)-1]} + ce^{-\beta r(x)}\right] + [1 - \eta(x)]\left[e^{\frac{\alpha}{2}[r(x)+1]} + ce^{-\beta r(x)}\right]$$
$$= e^{\frac{\alpha}{2}r(x)}I_\eta(x) + ce^{-\beta r(x)}.$$

Since the exponential function is convex, $\Delta\mathcal{C}_{\ell_1}(r, x)$ is a convex function of $r(x)$. Thus, for $r \in \mathcal{R}$, we obtain the minimum $r_0(x)$ by differentiating with respect to $r(x)$ and setting to 0:

$$\frac{\alpha}{2}e^{\frac{\alpha}{2}r(x)}I_\eta(x) - \beta ce^{-\beta r(x)} = 0$$

$$\Leftrightarrow r_0(x) = \log\left[\left(\frac{2\beta c}{\alpha I_\eta(x)}\right)^{\frac{2}{2\beta+\alpha}}\right].$$

Note that for all $x \in \mathcal{X}$, $\{r(x): r \in \mathcal{R}\} = [-\bar{r}(x), \bar{r}(x)]$. If $r_0(x)$ is within this range, plugging in $r_0(x)$ in $\mathcal{C}_{\ell_1}$ gives the corresponding minimal calibration gap $\mathcal{C}^*_{\ell_1}(x)$: $\mathcal{C}^*_{\ell_1}(x) = \left[\left(\frac{2\beta c}{\alpha}\right)^\gamma\right]I_\eta(x)^{1-\gamma}\left(\frac{1}{1-\gamma}\right)$. Otherwise, the corresponding minimal calibration gap is achieved at

either $r(x) = \overline{r}(x)$ or $r(x) = -\overline{r}(x)$. Plugging in these expressions give the corresponding minimal calibration gap $\mathcal{C}^*_{\ell_1}(x)$:

$$\mathcal{C}^*_{\ell_1}(x) = \begin{cases} \left[\left(\frac{2\beta c}{\alpha}\right)^\gamma\right] I_\eta(x)^{1-\gamma}\left(\frac{1}{1-\gamma}\right) & -\overline{r}(x) \le r_0(x) \le \overline{r}(x) \\ e^{\frac{\alpha}{2}\overline{r}(x)} I_\eta(x) + c e^{-\beta \overline{r}(x)} & r_0(x) > \overline{r}(x) \\ e^{-\frac{\alpha}{2}\overline{r}(x)} I_\eta(x) + c e^{\beta \overline{r}(x)} & r_0(x) < -\overline{r}(x). \end{cases}$$

This completes the proof. $\qquad\square$

## D.4 $\mathcal{H}$-CONSISTENCY BOUND

In this section, we prove our main result. The following result extends Proposition 4 to any hypothesis set $\mathcal{R}$ that is *regular for rejection* and will provide a key tool to derive our result. The proof is nearly identical to Proposition 4.

**Proposition 12.** *Assume that $\mathcal{R}$ is regular for rejection. Assume that there exists a convex function $\Psi: \mathbb{R}_+ \to \mathbb{R}$ with $\Psi(0) = 0$ such that the following holds for all $r \in \mathcal{R}$ and $x \in \mathcal{X}$:*
$\Psi\big(|\eta(x) - (1-c)|\mathbb{I}_{r(x)r^*(x)\le 0}\big) \le \Delta\mathcal{C}_{\ell_1}(0, x)$. *Let $\overline{I}_c$ be defined by $\overline{I}_c = c e^{\frac{\alpha}{2}} + (1-c) e^{-\frac{\alpha}{2}}$ and assume that $\frac{2\beta c}{\alpha} = \overline{I}_c$. Then, for any $r \in \mathcal{R}$:*

$$\Psi\big(R_{\ell_2}(r) - R^*_{\ell_2,\mathcal{R}} + \mathcal{M}_{\ell_2,\mathcal{R}}\big) \le R_{\ell_1}(r) - R^*_{\ell_1,\mathcal{R}} + \mathcal{M}_{\ell_1,\mathcal{R}}. \tag{6}$$

*Proof.* We will show that the following holds: $\inf_{r(x)r^*(x)\le 0}\Delta\mathcal{C}_{\ell_1}(r, x) = \Delta\mathcal{C}_{\ell_1}(0, x)$. The result then follows by Theorem 1 and Lemma 10. Since we have $\frac{2\beta c}{\alpha} = \overline{I}_c$, the following equivalence holds:

$$\begin{aligned} r_0(x) > 0 &\Leftrightarrow \frac{2\beta c}{\alpha I_\eta(x)} > 1 \\ &\Leftrightarrow I_\eta(x) < \overline{I}_c \\ &\Leftrightarrow \eta(x) > \frac{e^{\frac{\alpha}{2}} - \overline{I}_c}{e^{\frac{\alpha}{2}} - e^{-\frac{\alpha}{2}}} \\ &\Leftrightarrow \eta(x) > \frac{(1-c)e^{\frac{\alpha}{2}} - (1-c)e^{-\frac{\alpha}{2}}}{e^{\frac{\alpha}{2}} - e^{-\frac{\alpha}{2}}} \\ &\Leftrightarrow r^*(x) > 0. \end{aligned}$$

This implies $\inf_{r(x)r^*(x)\le 0}\mathcal{C}_{\ell_1}(r, x) = \inf_{r(x)r_0(x)\le 0}\mathcal{C}_{\ell_1}(r, x)$. Now, since $r_0(x)$ is the unique minimizer of the strictly convex function $\mathcal{C}_{\ell_1}(r, x)$ of $r(x)$, then, as a function of $r(x)$, $\mathcal{C}_{\ell_1}(r, x)$ is decreasing from $-\infty$ to $r_0(x)$ and increasing from there to $+\infty$. Thus, if $r_0(x) > 0$, the infimum of $\mathcal{C}_{\ell_1}(r, x)$ over $r(x) \le 0$ is reached for $r(x) = 0$. Similarly, if $r_0(x) < 0$, the infimum of $\mathcal{C}_{\ell_1}(r, x)$ over $r(x) \ge 0$ is reached for $r(x) = 0$. This shows that $\inf_{r(x)r_0(x)\le 0}\mathcal{C}_{\ell_1}(r, x) = \mathcal{C}_{\ell_1}(0, x)$, and completes the proof. $\qquad\square$

The following is our main result; it relates the surrogate estimation error to that of the rejection loss.

**Theorem 7.** *Assume that $\mathcal{R}$ is bounded with function $\overline{r}: \mathcal{X} \to \mathbb{R}$. Let $\alpha, \beta > 0$ be such that $\frac{2\beta c}{\alpha} = \overline{I}_c$, where $\overline{I}_c = c e^{\frac{\alpha}{2}} + (1-c) e^{-\frac{\alpha}{2}}$. Then, the following inequality holds for any $r \in \mathcal{R}$:*

$$R_{\ell_2}(r) - R^*_{\ell_2,\mathcal{R}} + \mathcal{M}_{\ell_2,\mathcal{R}} \le \Gamma\big(R_{\ell_1}(r) - R^*_{\ell_1,\mathcal{R}} + \mathcal{M}_{\ell_1,\mathcal{R}}\big), \tag{4}$$

*where* $\Gamma(z) = \begin{cases} \frac{2}{e^{\frac{\alpha}{2}} - e^{-\frac{\alpha}{2}}}\sqrt{\frac{(c+\overline{I}_c)\overline{I}_c}{c}}\, z & 0 \le z \le \frac{1}{4}\frac{c\overline{I}_c}{c+\overline{I}_c}\left[1 - e^{-\frac{\alpha(\overline{I}_c+c)\inf_{x\in\mathcal{X}}\overline{r}(x)}{2c}}\right]^2 \\ \frac{4(c+\overline{I}_c)}{c\left[1 - e^{-\frac{\alpha(\overline{I}_c+c)\inf_{x\in\mathcal{X}}\overline{r}(x)}{2c}}\right]\left[e^{\frac{\alpha}{2}} - e^{-\frac{\alpha}{2}}\right]}\, z & \text{otherwise.} \end{cases}$

*Proof.* Using the expression of $\Delta \mathcal{C}_{\ell_1}$ given by Lemma 11, we can write

$$\Delta \mathcal{C}_{\ell_1}(0,x) = \begin{cases} I_\eta(x) + c - \frac{1}{1-\gamma}\left(\frac{2\beta c}{\alpha}\right)^\gamma I_\eta(x)^{1-\gamma} & -\overline{r}(x) \le r_0(x) \le \overline{r}(x) \\ I_\eta(x) + c - e^{\frac{\alpha}{2}\overline{r}(x)} I_\eta(x) - ce^{-\beta\overline{r}(x)} & r_0(x) > \overline{r}(x) \\ I_\eta(x) + c - e^{-\frac{\alpha}{2}\overline{r}(x)} I_\eta(x) - ce^{\beta\overline{r}(x)} & r_0(x) < -\overline{r}(x). \end{cases}$$

$$= \begin{cases} I_\eta(x) + c - \left(\overline{I}_c + c\right)\left(\frac{I_\eta(x)}{\overline{I}_c}\right)^{1-\gamma} & -\overline{r}(x) \le r_0(x) \le \overline{r}(x) \\ I_\eta(x) + c - e^{\frac{\alpha}{2}\overline{r}(x)} I_\eta(x) - ce^{-\frac{\alpha \overline{I}_c \overline{r}(x)}{2c}} & r_0(x) > \overline{r}(x) \\ I_\eta(x) + c - e^{-\frac{\alpha}{2}\overline{r}(x)} I_\eta(x) - ce^{\frac{\alpha \overline{I}_c \overline{r}(x)}{2c}} & r_0(x) < -\overline{r}(x). \end{cases}$$

Without loss of generality, we consider $r^*(x) = \eta(x) - (1-c) \ge 0$. Then $r_0(x) \ge 0$. As with the proof of Theorem 5, we can express $\Delta \mathcal{C}_{\ell_1}(0,x)$ in terms of $u(x) = \eta(x) - (1-c)$, using $I_\eta(x) = J_u(x) + \overline{I}_c$, with $J_u(x) = \left[e^{-\frac{\alpha}{2}} - e^{\frac{\alpha}{2}}\right]u(x)$. Note that the condition $r_0(x) \le \overline{r}(x)$ can be expressed as

$$\log\left[\left(\frac{2\beta c}{\alpha I_\eta(x)}\right)^{\frac{2}{2\beta+\alpha}}\right] \le \overline{r}(x) \iff u(x) \le \frac{\overline{I}_c\left[1 - e^{-\frac{\alpha(\overline{I}_c+c)\overline{r}(x)}{2c}}\right]}{e^{\frac{\alpha}{2}} - e^{-\frac{\alpha}{2}}}.$$

When $0 \le r_0(x) \le \overline{r}(x)$, we have

$$\Delta \mathcal{C}_{\ell_1}(0,x) \ge \frac{1}{4}\frac{c\overline{I}_c}{c+\overline{I}_c}\left[\frac{J_u(x)}{\overline{I}_c}\right]^2 = \frac{1}{4}\frac{c\overline{I}_c}{c+\overline{I}_c}\left[\frac{u(x)}{\overline{I}_c}\right]^2\left[e^{\frac{\alpha}{2}} - e^{-\frac{\alpha}{2}}\right]^2.$$

When $r_0(x) > \overline{r}(x)$, we have

$$\Delta \mathcal{C}_{\ell_1}(0,x) \ge \frac{1}{4}\frac{c\overline{I}_c}{c+\overline{I}_c}\frac{\left[1 - e^{-\frac{\alpha(\overline{I}_c+c)\overline{r}(x)}{2c}}\right]}{\overline{I}_c}\left[e^{\frac{\alpha}{2}} - e^{-\frac{\alpha}{2}}\right]u(x).$$

Therefore, the function $\Psi(u)$ defined by

$$\Psi(u) = \begin{cases} \frac{1}{4}\frac{c\overline{I}_c}{c+\overline{I}_c}\left[\frac{u(x)}{\overline{I}_c}\right]^2\left[e^{\frac{\alpha}{2}} - e^{-\frac{\alpha}{2}}\right]^2 & 0 \le u(x) \le \frac{\overline{I}_c\left[1 - e^{-\frac{\alpha(\overline{I}_c+c)\inf_{x\in\mathcal{X}}\overline{r}(x)}{2c}}\right]}{e^{\frac{\alpha}{2}} - e^{-\frac{\alpha}{2}}} \\ \frac{1}{4}\frac{c\overline{I}_c}{c+\overline{I}_c}\frac{\left[1 - e^{-\frac{\alpha(\overline{I}_c+c)\inf_{x\in\mathcal{X}}\overline{r}(x)}{2c}}\right]}{\overline{I}_c}\left[e^{\frac{\alpha}{2}} - e^{-\frac{\alpha}{2}}\right]u(x) & \text{otherwise} \end{cases}$$

verifies the condition of Proposition 12 and therefore we have $\Psi\left(R_{\ell_2}(h) - R_{\ell_2}^*\right) \le R_{\ell_1}(h) - R_{\ell_1}^*$. An explicit upper-bound on $R_{\ell_2}(h) - R_{\ell_2}^*$ can be written in terms of $\Psi^{-1}$: $R_{\ell_2}(h) - R_{\ell_2}^* \le \Psi^{-1}\left(R_{\ell_1}(h) - R_{\ell_1}^*\right)$. To derive the expression of $\Psi^{-1}$, we write $z = \Psi(u)$, that is: when $0 \le z \le \frac{1}{4}\frac{c\overline{I}_c}{c+\overline{I}_c}\left[1 - e^{-\frac{\alpha(\overline{I}_c+c)\inf_{x\in\mathcal{X}}\overline{r}(x)}{2c}}\right]^2$,

$$4\frac{c+\overline{I}_c}{c\overline{I}_c}z = \left[\frac{u(x)}{\overline{I}_c}\right]^2\left[e^{\frac{\alpha}{2}} - e^{-\frac{\alpha}{2}}\right]^2 \iff |u| = \frac{2}{e^{\frac{\alpha}{2}} - e^{-\frac{\alpha}{2}}}\sqrt{\frac{(c+\overline{I}_c)\overline{I}_c}{c}z}.$$

Otherwise,

$$z = \frac{1}{4}\frac{c\overline{I}_c}{c+\overline{I}_c}\frac{\left[1 - e^{-\frac{\alpha(\overline{I}_c+c)\inf_{x\in\mathcal{X}}\overline{r}(x)}{2c}}\right]}{\overline{I}_c}\left[e^{\frac{\alpha}{2}} - e^{-\frac{\alpha}{2}}\right]u(x) \iff u = \frac{4(c+\overline{I}_c)}{c\left[1 - e^{-\frac{\alpha(\overline{I}_c+c)\inf_{x\in\mathcal{X}}\overline{r}(x)}{2c}}\right]\left[e^{\frac{\alpha}{2}} - e^{-\frac{\alpha}{2}}\right]}z$$

Thus, we have, for all $r \in \mathcal{R}$, $R_{\ell_2}(r) - R_{\ell_2}^* \le \Gamma\left(R_{\ell_1}(r) - R_{\ell_1}^*\right)$, where

$$\Gamma(z) = \begin{cases} \frac{2}{e^{\frac{\alpha}{2}} - e^{-\frac{\alpha}{2}}}\sqrt{\frac{(c+\overline{I}_c)\overline{I}_c}{c}z} & 0 \le z \le \frac{1}{4}\frac{c\overline{I}_c}{c+\overline{I}_c}\left[1 - e^{-\frac{\alpha(\overline{I}_c+c)\inf_{x\in\mathcal{X}}\overline{r}(x)}{2c}}\right]^2 \\ \frac{4(c+\overline{I}_c)}{c\left[1 - e^{-\frac{\alpha(\overline{I}_c+c)\inf_{x\in\mathcal{X}}\overline{r}(x)}{2c}}\right]\left[e^{\frac{\alpha}{2}} - e^{-\frac{\alpha}{2}}\right]}z & \text{otherwise.} \end{cases}$$

$\square$

Theorem 7 implies the following corollary for $\mathcal{R} = \mathcal{R}_{\text{lin}}$ and $\mathcal{R}_{\text{NN}}$.

**Corollary 8.** *Let $\mathcal{R} = \mathcal{R}_{\text{lin}}$ or $\mathcal{R}_{\text{NN}}$. Let $\alpha, \beta > 0$ be such that $\frac{2\beta c}{\alpha} = \bar{I}_c$, where $\bar{I}_c = ce^{\frac{\alpha}{2}} + (1-c)e^{-\frac{\alpha}{2}}$. Then, the following inequality holds for any $r \in \mathcal{R}$:*

$$R_{\ell_2}(r) - R^*_{\ell_2, \mathcal{R}} + \mathcal{M}_{\ell_2, \mathcal{R}} \le \Gamma\big(R_{\ell_1}(r) - R^*_{\ell_1, \mathcal{R}} + \mathcal{M}_{\ell_1, \mathcal{R}}\big), \tag{5}$$

*where $\Gamma(z) = \begin{cases} \frac{2}{e^{\frac{\alpha}{2}} - e^{-\frac{\alpha}{2}}} \sqrt{\frac{(c + \bar{I}_c)\bar{I}_c}{c}} z & 0 \le z \le \frac{1}{4} \frac{c\bar{I}_c}{c + \bar{I}_c}\left[1 - e^{-\frac{\alpha(\bar{I}_c + c)B}{2c}}\right]^2 \\ \frac{4(c + \bar{I}_c)}{c\left[1 - e^{-\frac{\alpha(\bar{I}_c + c)B}{2c}}\right]\left[e^{\frac{\alpha}{2}} - e^{-\frac{\alpha}{2}}\right]} z & \text{otherwise} \end{cases}$ and $B$ is replaced by $\Lambda B$ for $\mathcal{R} = \mathcal{R}_{\text{NN}}$.*

*Proof.* Using the fact that $\inf_{x \in \mathcal{X}} \bar{r}(x) = \inf_{x \in \mathcal{X}}\big(W\|x\|_p + B = B\big)$ for $\mathcal{R} = \mathcal{R}_{\text{lin}}$ and $\inf_{x \in \mathcal{X}} \bar{r}(x) = \inf_{x \in \mathcal{X}}\big(\Lambda W\|x\|_p + \Lambda B\big) = \Lambda B$ for $\mathcal{R} = \mathcal{R}_{\text{NN}}$, by Theorem 7, we complete the proof. $\square$

