# OpenReview forum: "Learning to Reject with a Fixed Predictor: Application to Decontextualization"
_ICLR.cc/2024/Conference — ICLR 2024 poster_

### Official Review · Reviewer_irSz · 2023-10-29

**Soundness:** 3 good
**Presentation:** 2 fair
**Contribution:** 2 fair
**Rating:** 6
**Confidence:** 3

**Summary:**

In the submission, a novel loss function parameterised by the learnable rejectors is studied to train the LLM model to handle diverse outputs for one prompt. To compute the existing loss function for tackling this problem is NP-hard. At the same time, the authors proposed a trackable surrogate loss, which is differentiable and convex, and by optimising it, the generalisation error is minimised supported by the theorem proposed as the main result in the submission. In the experiments, by training the model with the proposed loss function, the models are improved in terms of precision vs. Coverage compared with the existing method.

**Strengths:**

1. The derivation is clear and sound. I am not an expert on the proofing behind so I only scan the appendix. However, by following each proposition and theorem, it is easy to follow the logic step by step and reach the point that by minimising the proposed loss function the target generalisation error is minimised.
2. Nice plot for intuitively explaining the property of the surrogate loss with the changing of the r(x).

**Weaknesses:**

1. The experiment setting is not well explained and the settings are not comprehensive. As the authors mentioned,  the LLM predictors are their main focus but in the experiments, there is only one type of LLM. As T5X is a family of models, there are other choices and more architectures from other families will be more comprehensive for evaluating the proposed loss.
2. The loss function is tested on image classification but on a tiny setting and this pure classification task does not really relate to the LLM setting.
3. To train the rejector still required to label new information, it is hard to distinguish whether the improvement is from the additional information or the loss function.

**Questions:**

1. According to my understanding, applying the surrogate loss requires labelling the output from the given model. Then the model is further trained by the learned surrogate loss. Thus some extra information is introduced. Can the Cross entropy loss and Maxprob have the same information?
2. What is the format of the ejector?
3. WHy the F1 score is not applied for clear comparison?
5. For the std comparison is not very clear whether rejection loss is much better than Maxprob in Figure 4 and in Figure 5.

---

> ### Author Response · Authors · 2023-11-14
>
> Thank you for your thoughtful feedback. We have carefully addressed all the questions raised. Please find our responses below.
>
> **Weaknesses:**
>
> **1. The experiment setting is not well explained and the settings are not comprehensive. As the authors mentioned, the LLM predictors are their main focus but in the experiments, there is only one type of LLM. As T5X is a family of models, there are other choices and more architectures from other families will be more comprehensive for evaluating the proposed loss.**
>
> **Response:** We kindly ask the reviewer to elaborate on which part of the experiment setting was not sufficiently well explained. While it is true that there generally exist many LLM architectures, it would not be feasible to evaluate all of them. By showing very strong results on T5X XXL, a standard model, we demonstrate that there is at least one architecture for which our surrogate loss is very useful.
>
> **2. The loss function is tested on image classification but on a tiny setting and this pure classification task does not really relate to the LLM setting.**
>
> **Response:** Since annotation is expensive for the LLM setting, we chose to provide additional experimental evaluation of our proposed surrogate loss on image classification. For the LLM setting, please see our experiments on decontextualization.
>
> **3. To train the rejector still required to label new information, it is hard to distinguish whether the improvement is from the additional information or the loss function.**
>
> **Response:** Since the baseline using the cross-entropy loss had access to these new labels as well, the improvement is necessarily from the loss function.
>
> **Questions:**
>
> **1. According to my understanding, applying the surrogate loss requires labeling the output from the given model. Then the model is further trained by the learned surrogate loss. Thus some extra information is introduced. Can the Cross entropy loss and Maxprob have the same information?**
>
> **Response:** The cross entropy baseline did have the same information - we finetuned a model using the cross-entropy loss *on the annotations*. However, it would not be possible for Maxprob to also benefit from this information because the annotations form a binary classification task, while the predictor (from which Maxprob is derived) is trained to generate decontexualizations.
>
> **2. What is the format of the rejector?**
>
> **Response:** The format for the rejector is described in section 7.2 (surrogate loss), and it is trained on the decontextualization rejection task described in section 7.1. At a high level, its input is a sentence and a candidate decontextualization, and its output is a real number whose sign corresponds to acceptance or rejection.
>
> **3. Why is the F1 score not applied for clear comparison?**
>
> **Response:** The F1 score relates to precision versus recall, while instead we are interested in precision versus coverage. We believe that it is very natural and easy to compare the methods by plotting precision versus coverage (which we do in Figures 4, 5, and 6).
>
> **4. For the std comparison it is not very clear whether rejection loss is much better than Maxprob in Figure 4 and in Figure 5.**
>
> **Response:** We respectfully disagree with the reviewer. The curve for the rejection loss lies above that of Maxprob in both figures, and in Figure 4, it lies *significantly* above that of Maxprob.

---

> > ### Author Response · Authors · 2023-11-22
> >
> > Dear Reviewer, we sincerely thank you once again for your thoughtful feedback and valuable comments. As the Author/Reviewer Discussion phase draws to a close, we kindly request that you let us know if our response has convincingly shown that the improvement in empirical results is indeed attributable to our new loss function. If you have any additional questions or comments, we are happy to respond to them.

---

### Official Review · Reviewer_KRjg · 2023-10-31

**Soundness:** 3 good
**Presentation:** 3 good
**Contribution:** 3 good
**Rating:** 6
**Confidence:** 3

**Summary:**

This paper studies the problem of learning to reject with a fixed predictor, motivated by the case when the fixed predictor is a pretrained large language model (LLM). The goal is to learn a rejector function that allows for post-hoc filtering of LLM outputs: by rejecting lower-quality outputs, the combined predictor/rejector system can have higher precision at the cost of lower coverage.
This is especially critical in high-stakes sequence-to-sequence problems in domains such as health data.

The paper designs an H-consistent surrogate loss for learning to reject with a fixed predictor using the framework of Awasthi et al. (2022). This loss is then used for the decontextualization problem, which is to rephrase a sentence into a form that's understandable without the context around the original sentence.

On the decontextualization experiment, the proposed surrogate outperforms strong baselines from the learning-to-reject and model calibration literature.

**Strengths:**

- The surrogate loss derived in the paper doesn't require tuning of a threshold parameter, which is a drawback of confidence-based approaches.

- Empirical evaluation that avoids a common pitfalls of other work in this area: because different loss functions are being compared, it's only fair to compare performance with the optimal learning rate, since changing the loss function changes the scale of gradient updates.

- The paper's theoretically-derived relationship between the two hyperparameters $\alpha$ and $\beta$ works well empirically.

**Weaknesses:**

- The method is only applied to the decontextualization problem when it actually seems to have much more broad applicability. I can think of several LLM applications where the ability to increase precision by rejecting would be useful. For example, we could try to learn a rejector that cuts down on hallucinations in summarization or text simplification. More exploration of this technique beyond the decontextualization problem would make it more impactful.

- > Additionally, to the best of our knowledge, minimizing the cross-entropy loss does not have any proven guarantee with respect to our main objective: minimizing the induced rejection loss.

    - When $\mathcal{R} = \mathcal{R}\_{all}$, can't we use a cost-sensitive surrogate loss plus usual (Bayes) consistency results?
      I.e., I don't understand the following claim in Appendix C given that the paper's results only consider $\mathcal{R} = \mathcal{R}\_{all}$:
     >  (i) There is a lack of any H-consistency bound guarantees for cost-sensitive surrogate losses with respect to the induced rejection loss.

        In general, I think the paper could use more presentation on why the naive approach of directly training the rejector to predict the labels $a$ using a standard cost-sensitive surrogate loss doesn't work, since the results only consider the $\mathcal{R} = \mathcal{R}\_{all}$ case. That, or some results for linear or 1-layer-NN rejectors, as in Awasthi et al. (2022), would strengthen the theoretical part of the paper.

- No ablation on $\alpha$ even though it's important in the bound (Thm 5), and different experiments use different values (e.g., 4 in main text experiments, 3.5 in appendix vision experiments)

**Questions:**

- Why do we care about $\mathcal{H}$-consistency even though we only consider the space $\mathcal{R}_{all}$ of all measurable functions? It would be helpful to further emphasize this earlier in the paper as a major difference / novelty / reason why more naive baselines don't work (see earlier comment under weaknesses).

- small typo in display at bottom of pg4; $\le 0$ should be in the subscript of $\mathbb{I}$

- small typo in display at bottom of pg4; not enough parentheses

- Minor semantics/notation: $r(x,f(x))$ reads like a score for rejecting, so the notation reads like $r>0 \implies$ reject, when it's actually being used in the opposite sense. This tripped me up several times. I know this notation is inherited from Cortes et al. but I think it's a bit confusing.

---

> ### Author Response · Authors · 2023-11-14
>
> Thank you for your encouraging review. We will take your suggestions into account when preparing the final version. Below please find responses to specific questions.
>
> **1. The method is only applied to the decontextualization problem when it actually seems to have much more broad applicability. I can think of several LLM applications where the ability to increase precision by rejecting would be useful. For example, we could try to learn a rejector that cuts down on hallucinations in summarization or text simplification. More exploration of this technique beyond the decontextualization problem would make it more impactful.**
>
> **Response:** As discussed in the conclusion, annotation for the LLM setting is expensive and this limits our empirical evaluation. We therefore provided additional experimental evaluation on image classification tasks, where checking the correctness of the predictor is easy. We leave further experimental evaluation for future work, but observe very strong results on our current LLM experiments.
>
> **2. When $\mathcal{R} =
> R_{\mathrm{all}}$, can't we use a cost-sensitive surrogate loss plus usual (Bayes) consistency results? I.e., I don't understand the following claim in Appendix C given that the paper's results only consider  $\mathcal{R} = \mathcal{R}_{\mathrm{all}}$.**
>
> **In general, I think the paper could use more presentation on why the naive approach of directly training the rejector to predict the labels $a$ using a standard cost-sensitive surrogate loss doesn't work, since the results only consider the $\mathcal{R} = \mathcal{R}_{\mathrm{all}}$ case. That, or some results for linear or 1-layer-NN rejectors, as in Awasthi et al. (2022), would strengthen the theoretical part of the paper.**
>
> **Question 1. Why do we care about $\mathcal{H}$-consistency even though we only consider the space $\mathcal{R}_{\mathrm{all}}$ of all measurable functions? It would be helpful to further emphasize this earlier in the paper as a major difference / novelty / reason why more naive baselines don't work (see earlier comment under weaknesses).**
>
> **Response:** Our surrogate losses benefit from $\mathcal{R}$-consistency bounds with the hypothesis set $\mathcal{R}$ extending beyond the family of all measurable functions $\mathcal{R}_{\mathrm{all}}$. This includes, for example, the family of linear models and one-hidden-layer neural networks considered by Awasthi et al. (2022). For simplicity, we presented results only for the family of all measurable functions. However, our proofs and results can be directly extended to other hypothesis sets. In the final version, we will include these $\mathcal{R}$-consistency bounds.
>
> While the cost-sensitive surrogate loss benefits from Bayes-consistency, it does not benefit from $\mathcal{R}$-consistency bounds. As pointed out by Awasthi et al. (2022), Bayes-consistency does not supply any information about learning with a typically restricted hypothesis set, which of course would not contain all measurable functions. Furthermore, it provides no guarantee for approximate minimizers since the rate of that asymptotic convergence could be arbitrarily slow.
>
> **3. No ablation on $\alpha$ even though it's important in the bound (Thm 5), and different experiments use different values (e.g., 4 in main text experiments, 3.5 in appendix vision experiments).**
>
> **Response:** In our experiments on image classification, the value of $\alpha$ is selected using cross-validation, a method typically applicable in practice. In our future work, we aim to explore a more principled approach for choosing $\alpha$. A promising direction involves establishing data-dependent bounds, which would act as a useful guide for selecting $\alpha$ across different datasets.
>
> **4. Small typo in display at bottom of pg4; $\leq 0$ should be in the subscript of $\mathbb{I}$.**
>
> **Response:** Thank you, we will fix this in the final version.
>
> **5. Small typo in display at bottom of pg4; not enough parentheses.**
>
> **Response:** Thank you, we will fix this in the final version.
>
> **6. Minor semantics/notation: $r(x, f(x))$ reads like a score for rejecting, so the notation reads like $r > 0$ $ \implies $ reject, when it's actually being used in the opposite sense. This tripped me up several times. I know this notation is inherited from Cortes et al. but I think it's a bit confusing.**
>
> **Response:** We are indeed staying consistent with the notation from the literature.

---

> ### Author Response · Authors · 2023-11-21
>
> We have uploaded a revised version with a newly added Appendix D. The main results (on p.18), Theorem 7 and Corollary 8, demonstrate that our surrogate losses benefit from $\mathcal{R}$-consistency bounds with any bounded hypothesis set, including the family of linear models and one-hidden-layer ReLU networks considered in (Awasthi et al., 2022) as special cases. In the initial version, we mentioned before Section 5.5 "Similar results can be derived for other families of functions $\mathcal{R}$, such as that of linear functions or neural networks with one hidden-layer as in (Awasthi et al., 2022)", but did not include them for simplicity. We believe these newly added results will further strengthen the theoretical part of the paper and address the reviewer's concerns.
>
> Note that there is a lack of similar guarantees for cost-sensitive surrogate losses with respect to the induced rejection loss. Furthermore, our surrogate loss outperforms the cost-sensitive baseline in our image classification experiments.

---

> > ### Author Response · Authors · 2023-11-22
> >
> > Dear Reviewer, we sincerely thank you again for your insightful feedback and for appreciating our contribution. As the Author/Reviewer Discussion phase draws to a close, we would be grateful if you could let us know whether our response, along with the full appendix, has sufficiently addressed your concerns.

---

> > > ### Comment · Reviewer_KRjg · 2023-11-22
> > > **update**
> > >
> > > > While the cost-sensitive surrogate loss benefits from Bayes-consistency, it does not benefit from $\mathcal{R}$-consistency bounds
> > >
> > > I agree---my point was that when $\mathcal{R} = \mathcal{R}\_{all}$, as it is for your main-text results, the Bayes-consistentcy results for usual cost-sensitive surrogates should directly apply. That makes including results for $\mathcal{R} \ne \mathcal{R}_{all}$ crucial for your novelty argument compared to other losses. Since you added such results to the appendix, I will increase my score.
> > >
> > > I still the the paper would be much improved by:
> > > - (1) including some $\mathcal{R} \ne \mathcal{R}_{all}$ results in the *main text* to emphasize a key difference between the proposed surrogate and others, and additional high-level discussion of why $\mathcal{R}$-consistency matters to make it clear for an NLP audience.
> > > - (2) LLM experiments on problems beyond decontextualization. E.g., does this technique reduce hallucinations for summarization while maintaining good performance on the not-rejected subset? You can use fully automated annotation metrics, e.g. by measuring hallucination as long spans that don't appear in the input text. What's the pareto curve of hallucination vs rejection fraction? Etc.

---

> > > > ### Author Response · Authors · 2023-11-22
> > > >
> > > > We thank the reviewer for their response and for increasing the score. We will incorporate your suggestions to enhance the paper in its final version. This will include adding general $\mathcal{R}$-consistency bounds in the main text, accompanied by a more comprehensive high-level discussion. We will also seek to include experimental results beyond decontextualization. If the reviewer could recommend a specific summarization task or dataset, we would greatly appreciate it.

---

> > > > > ### Comment · Reviewer_KRjg · 2023-11-22
> > > > > **summarization**
> > > > >
> > > > > This might not be the best choice, but it could be a good jumping-off point: https://aclanthology.org/2022.acl-long.236.pdf
> > > > > This is a summary dataset with hand-labeled entities extracted from the summaries. The entities are labeled as factual / non-factual (hallucinated). They use this to train a detector for hallucinated entities which you could then apply to your test set predictions. Can the proposed approach decrease the rate of hallucinated entities while maintaining summarization quality?

---

> > > > > > ### Author Response · Authors · 2023-11-22
> > > > > > **Response to summarization**
> > > > > >
> > > > > > We thank the reviewer for proposing this dataset. It indeed seems applicable to our scenario, and we will thus try to report experimental results for the final version. Given our decontextualization experiment, we anticipate that our proposed method will effectively reject the hallucinated entities.

---

### Official Review · Reviewer_AHeX · 2023-11-03

**Soundness:** 3 good
**Presentation:** 3 good
**Contribution:** 3 good
**Rating:** 6
**Confidence:** 3

**Summary:**

For the purpose of letting a model 'reject' or 'abstain' from classifying some samples, a technique of using a rejection loss is proposed; from which a surrogate loss is derived and used. This is in contrast to either using the 'confidence' output or training for an (n+1)-th class in the sense that the predictor can be fixed. Theoretical guarantees for this surrogate loss are provided, and this framework is evaluated on real-world decontextualization tasks, and also for image classification. The results show promise and seem to perform better than other methods studied.

**Strengths:**

- Very promising real-world results
- Theoretical guarantees motivated well and provided
- Good theoretical comparison to other methods, and good motivation for the need of proposed method provided
- NLP examples but possibly further extensions to other areas
- Surrogate loss performance better than other models and also quite close to theoretical limits at times.

**Weaknesses:**

- Although GitHub repo links and other identifying information cannot be written in a paper under review, I did not see any indication of the intention to make the code public, nor is it provided in supplementary materials
- Page 4. First equation/inequality.
	- $c$ is positive. indicator functions are either 0 or 1, so:
$\mathbb{I}\_{a\leq 0} \mathbb{I}\_{r(x)>0} + c \mathbb{I}\_{r(x)\leq 0} \geq max(\mathbb{I}\_{a\leq 0}\mathbb{I}\_{-r(x)<0}, c \mathbb{I}\_{r(x)\leq 0})$

	max of two different terms that are positive should be less than or equal to the sum.
	 - And second comparison should be equal. As the first term in the first max is saying: "Both $a$ and $-r$ should be less than zero for the indicator product to be one". And the the first term in the second max is saying "the max of both $a$ and $-r$ should be less than zero for the indicator to be one". Both of these statements imply each other and therefore the last relation should be of equality.
	 - The bound becomes a lower bound, not an upper bound.
	 - I did not check the last relation. That might still hold despite this, but need to know why.
 - There is no test set. Only train and validation, where cross-validation is used so training algorithm sees all data.

**Questions:**

- page 5. it is said that "underlying scores are not favourable for that precision level". Why is that?
- These are possibly standard deviation bars in figures 4 and 5. How were they generated? Is it from different folds of cross-validation?
- Are Maxprob and cross-entropy trained on different models? Why is that?

**Minor Typing / Formatting / Clarity issues**
- page 4. last equation "<=0" should be in the subscript
- please recheck format for citations: some citations use et al while other list all authors.

**Comment**: I chose "good" in soundness, presentation and contribution, but "3: reject" in the overall rating. That's mainly because of the mathematical inconsistency, which I hope can be resolved.

---

> ### Comment · Reviewer_KRjg · 2023-11-10
> **re: inequality on pg 4**
>
> Note that one term in the sum has $\mathbb{I}[r > 0]$ and the other term has $\mathbb{I}[r \le 0]$, so only one term can be nonzero. This yields the first inequality.

---

> ### Author Response · Authors · 2023-11-14
>
> Thank you for your appreciation of our contribution. We have carefully addressed all the questions raised, especially regarding the consistency of our mathematical results. Please find our responses below, and we respectfully request the reviewer to reconsider the overall rating.
>
> **Weaknesses:**
>
> **1. Although GitHub repo links and other identifying information cannot be written in a paper under review, I did not see any indication of the intention to make the code public, nor is it provided in supplementary materials.**
>
> **Response:** We are hoping to release code for the final version and are waiting for approval.
>
> **2. Page 4. First equation/inequality.**
>
> **Response:** As pointed out by reviewer KRjg, the first inequality indeed holds. Thus, we are still giving an upper bound. We will replace the second inequality with equality (we were simply staying consistent with Cortes et al.). The last relation in the derivation holds because a max of two positive real numbers is upper-bounded by their sum.
>
> **3. There is no test set. Only train and validation, where cross-validation is used so training algorithm sees all data.**
>
> **Response:** We perform k-fold cross validation, and thus the training algorithm does not see the data it is tested on. Additionally, in the decontextualization experiment with LLMs, we do not tune any hyper-parameters for our proposed method.
>
> **Questions:**
>
> **1. page 5. it is said that "underlying scores are not favourable for that precision level". Why is that?**
>
> **Response:** We would like to point out that we have instead written “not *necessarily* favorable.” This is because the same scores are used for each precision level; the learning algorithm was not trained to target a specific precision level. On the other hand, in our surrogate loss, we are targeting a specific precision level through the value of $c$.
>
> **2. These are possibly standard deviation bars in figures 4 and 5. How were they generated? Is it from different folds of cross-validation?**
>
> **Response:** As described in their respective captions, the standard deviations in Figure 4 are from the different cross-validation splits, and the standard deviations in Figure 5 are from different training runs.
>
> **3. Are Maxprob and cross-entropy trained on different models? Why is that?**
>
> **Response:** These are very slightly different - Maxprob was derived from an existing decontextualization model, and we chose to use T5X for all of our fine-tuning as it was the closest model available to us.
>
> **Minor Typing / Formatting / Clarity issues:**
>
> **1. page 4. last equation "<=0" should be in the subscript**
>
> **2. please recheck format for citations: some citations use et al while other list all authors.**
>
> **Response:** Thank you, we will fix these in the final version.

---

> > ### Comment · Reviewer_AHeX · 2023-11-20
> > **Updating Review**
> >
> > Thank you for your response.
> >
> > **Inequality**
> > Thanks to reviewer KRjg for correcting me about the inequality. The second part can remain as an inequality if you want because that is also correct and follows the literature.
> >
> > **Error Bars**
> > I have no idea how I missed the description in the captions. Apologies for this oversight.
> >
> > **Cross Validation**
> > Section 7.1 says there are 1500 training and 500 validation examples. If it is k-fold cross validation then the fold is different each time and these 500 examples are not with-held then? Or is it that the k-fold cross validation is within the 1500 and the final testing is on the with-held 500 examples that the model does not see during training?
> >
> > I am updating my review.

---

> > > ### Author Response · Authors · 2023-11-21
> > > **Official Response by Authors**
> > >
> > > We thank the reviewer for their response and for updating the rating.
> > >
> > > For cross-validation, we do the former which we believe is standard. There were 4 non-intersecting folds of 500 validation samples, and we trained models on the remaining 1500. We also note again that we tuned the hyper-parameters for the baselines but not for our proposed method in the decontextualization experiment.

---

> > > > ### Comment · Reviewer_AHeX · 2023-11-22
> > > > **Cross Validation**
> > > >
> > > > Thank you for your response. Yes I believe that is indeed the standard, but that also means that the parameters are tuned on the validation set (when it becomes part of train set in all other three validation folds). Evem if the hyperparameters are not updated, the model parameters get updates. That is why I was hoping to see a test set. It does definitely seem like an important contribution despite the lack of a test set, but that was what my concern was.

---

### Meta-Review · Area_Chair_zsQ1 · 2023-12-02

**Metareview:**

This paper is focused on the selective classification setting where predictors are able to reject to improve precision at the cost of coverage. Specifically, upon introducing a new formulation for this setting, authors propose a convex surrogate loss to enable jointly learning a classifier and a rejection rule. A theoretical analysis is presented and an empirical assessment is carried out in both conditional text generation and image classification tasks.

The reviewers are consistent in highlighting the paper is well-written and tackles a relevant problem offering an efficient and simple solution. Our recommendation is thus for acceptance since we believe this paper will add significant value to the conference program.

I would like however to highlight that, as mentioned by some of the reviewers, the empirical assessment is limited and focuses only on one conditional generation task and one very simple image classification task. Future work should thus assess to what extent the proposal is beneficial in cases where other models, tasks, and datasets are considered. Moreover, The discussion in the paper suggests that the fact that the proposal is threshold-free is an improvement over alternative approaches that require a threshold to be decided upon during testing. The difference between those two groups of approaches are a bit more nuanced than one being clearly preferable than the other, and selecting a threshold gives the user the option to decide on precision and coverage levels without retraining. I'd recommend modifying that discussion a bit in the text to reflect that.

**Justification For Why Not Higher Score:**

The evaluation requires improvement to provide clear evidence of the authors's claims.

**Justification For Why Not Lower Score:**

The proposal is novel, relevant, and efficient.

---

### Decision · Program_Chairs · 2024-01-16

Accept (poster)